# Flora and Typology of Wetlands of Haho River Watershed, Togo

**Fousséni Folega [1,\*], Madjouma Kanda [1], Kossi Fandjinou [1], Eve Bohnett [2,\*], Kperkouma Wala [1], Komlan Batawila [1] and Koffi Akpagana [1]**

[1]  Géomatique et Modélisation des Écosystèmes, Laboratoire de Botanique et Écologie Végétale (LBEV), Département de Botanique, Faculté des Sciences, Université de Lomé, Lomé 01 BP 1515, Togo

[2]  Department of Biology, San Diego State University, San Diego, CA 92182, USA

\*  Correspondence: ffolegamez@live.fr (F.F.); evebohnett@yahoo.com (E.B.)

**Abstract:** Wetlands are recognized as hotspots of biodiversity and providers of several ecosystem services, including water purification, sediment stabilization, and flood, erosion, and climate regulation. This article aims to investigate the floristic diversity of the wetlands the Haho River watershed in southern Togo. Spatial data from Astrium service and Google Earth were collected, and phytosociological data were classified following the Braun–Blanquet approach. The findings indicate that 72 families in total have evolved in this environment, with Poaceae (14.95%) and Fabaceae (11.98%) dominating. The number of species was estimated to be 323; the three species that were most prevalent in the wetland's habitats were *Elaeis guineensis* Jacq (2.44%), *Panicum maximum* Jacq (2.29%), and *Lonchocarpus sericeus* (Poir) H. B. K. (1.71%). The most prevalent and abundant life forms in these moist habitats were micro-phanerophytes (34.70%) and therophytes (23.50%). However, the most common and abundant chorological categories included pantropical (31.05%) and Guinean-Congolese species (21.46%). Principal Component Analysis (PCA) was used to examine how abiotic parameters, including depth/degree of immersion, influence the distribution of plant species in a wetland landscape. This research has the potential to be developed into a more robust action study for wetland classification and recognition.

**Keywords:** biodiversity; wetlands; resilience; watershed; Togo

## 1. Introduction

During the last 150 years, 50% of the worldwide wetland habitats have vanished, and the remnants now cover only 6% of the land surface area [1,2]. Wetlands around the globe have been altered, degraded, or lost through a wide range of human activities altering their biological productivity [3,4]. These particular ecosystems link aquatic and terrestrial ecosystems as they provide specific ecosystem services, including water filtration, protection from ecological disturbances such as floods and hurricanes, support for food chains to maintain the biological cycle, and carbon sequestration [5,6]. Wetlands constitute an enormously diversified landscape component that spans a wide range of geographic locations and hydrographic networks in both aquatic (inland and coastal marshes) and terrestrial (lakes, pools, ponds, and dams) systems [7,8]. Wetlands are green infrastructures that serve as territorial resources and can deliver ecosystem services for the local population [8,9]. Due to their unique characteristics and complexities, these ecosystems are traditionally landscapes that need to be managed [10,11]. When wetlands are present in a metropolitan area experiencing economic growth and rapid population growth, policymakers have been especially concerned about planning to avoid degradation.

The production of wetland ecosystem knowledge, including the creation of various legal and technical tools, has advanced significantly in recent decades in the context of conservation and restoration programs on a global and regional scale [12]. Since the Ram-

sar Convention [13] on the recognition of wetlands, followed by the Convention on Biological Diversity [14], and the Millennium Ecosystem Assessment [15], there has been a resurgence in interest in the reclamation of wetlands for ecological, historical, and economic reasons. Therefore, the Ramsar treaty established precise principles/criteria for classifying wetland areas and drew increased local or regional attention to their sustainable monitoring. However, all methods of wetland appropriation require rational and multi-disciplinary reasoning [16,17]. The paradigm of reclassifying wetlands as structural and functional spatial units that constitute a crucial component of the territorial network was developed through multi-actor talks with stakeholders [17]. Wetland management has not always been approached by scientists, planners, and water managers in the same way [18], raising concerns about the value of biological variety and the impact of suitable management methods [19,20].

The rehabilitation and protection of wetlands throughout the world, particularly in developing countries, has been a source of social conflict; these conflicts have been accentuated by institutional weakness, followed by the lack of adequate knowledge of these critical ecosystems [21,22]. To effectively and efficiently manage wetland ecosystems, researchers agree that territorial sovereignty over wetlands must consider the local populations' participation and socio-economic resilience [23]. However, a significant barrier to the effective adoption of sustainable management strategies is the low degree of local government decentralization in the domain of wetlands.

In Togo, infrequent studies have shown interest in floodplains' biology to understand these ecosystems better. Studies have examined the hydrochemistry, phytosociology, and spatiotemporal dynamics of mangrove regions in the littoral belt [24]. Additionally, research has shown that *Sorindeia warneckei* Engl. has a positive economic impact on the people living in the Lama Depression [25].

However, little research has examined the floristic diversity, phytosociology, or production dynamics of wetlands at an eco-floristic or watershed scale; thus, the primary purpose of the present study is to fill this gap in our research area. Wetlands can contribute enormously to landscape-level biodiversity because of their species richness and spatial variation in community composition [26]. This article contributes to a greater understanding of Togo's wetland ecosystems and attracts the attention of natural resource managers interested in their sustainable management. Notably, South Togo's Haho River watershed's wetland's floristic diversity and ecological classification require in-depth analysis.

## 2. Materials and Methods

### 2.1. Study Area

The study area is a complex hydrological system, approximately 4165.62 km$^2$ (Figure 1), made up of rivers (Haho, Yoto, Lili, and Boko) and lakes (Togo and Zowla). The northernmost two-thirds are peneplains shared by Benin and Togo, typified by low elevations (below 200 m), but are also interspersed with hills and inselbergs. At a height of less than 100 m, the southern portion is a part of the coastal plain [27].

The area has a tropical Guinean climate, with 1000 and 1400 mm of rain yearly. Although it can occasionally reach extremes of 30 °C, the average temperature is around 27 °C. The majority of soils are shallowly leached tropical ferruginous types.

The study area is located in the coastal plain, the fifth phytogeographical domain V, and is characterized by a mosaic of severely degraded plant formations, including forest islands with *Milicia excelsa* (Welw.) C. C. Berg., *Antiaris africana* Engl., *Cola gigantea* A. Chev. Land cover is impacted by pedology and hydrography, which are the sources of physiognomic units, including agrarian landscapes, mining areas, swamps and water bodies, natural vegetation, plantations, and urban areas [28].

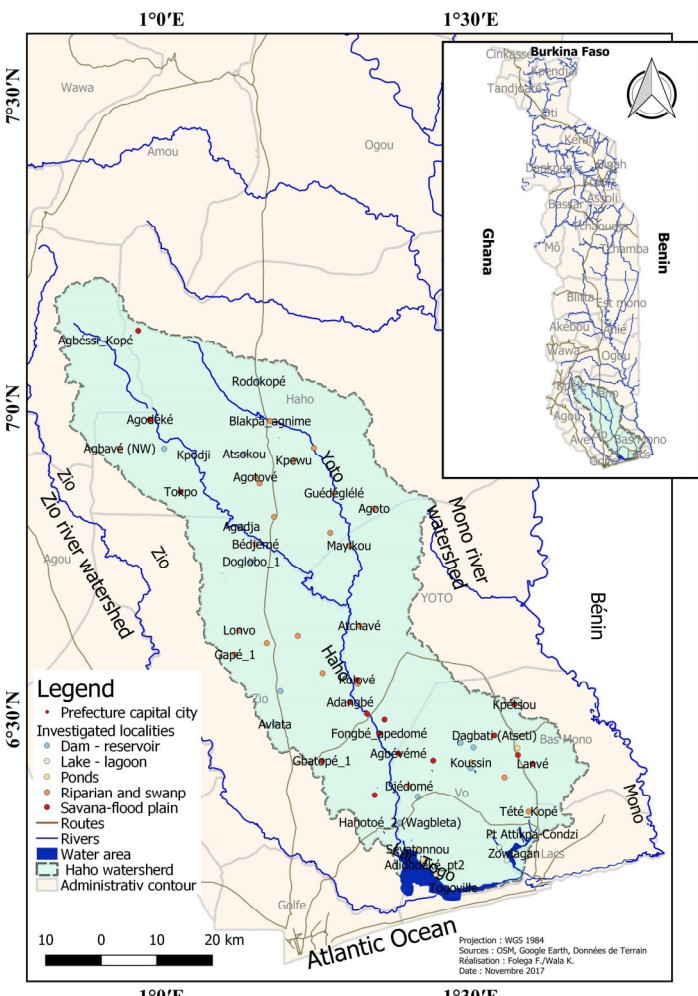

**Figure 1.** Location of wetlands sampled in the Haho basin (Togo).

The Adja, Ewé, Watchis, Pedas, Gins, and Minas [29] make up the key socio-cultural groups in a country with a population of over one million. Socio-economic development is supported by family-based agricultural production, cattle keeping, artisanal fishing, market gardening, small trading, and processing agricultural products. The lack of innovative capacity and inadequate technical support for rural development impacts yields and production performance in this area.

### 2.2. Data Collection

Identification and Geolocation of Wetlands in the Watershed

By comparing the spatial data (hydrographic, geomorphological, pedological, and vegetation maps) available, the wetlands to be surveyed were identified [30]. First, the geographical extent of the Haho River watershed could be defined using tools for visual analyses and interpretations, with geoprocessing in the QGIS software (Figure 1). Next, a KLM (Keyhole Markup Language) was created by geolocating the surface entities and delineating the observable bodies of water in the basin representing the potential wetlands to be studied. Before the floristic inventory was formed, the latter was put into Google Earth Pro to verify the efficacy and typology of wetlands. Sixty-four (64) sampling points were ultimately chosen after being compared to high-resolution photos that Astrium service had made available from Google Earth. The sampling points are divided into riparian forests (38), ponds and dams (15), around lakes and lagoons (6), and ponds

(5). It should be emphasized that riparian forests are complexes that include riparian forests and nearby marshland landforms. In general, there are two main types of wetlands: those that are natural (such as riparian forests, lakes, lagoons, and ponds) and those that are constructed (ponds and dams).

*2.3. Floristic and Ecological Inventories of Wetlands*

The phytosociological data of the wetlands in our sample were collected following the sigmatist approach of Braun–Blanquet [30]. The floristic surveys were conducted in 100 m² plots [31,32]. This survey method is appropriate for examining formations that are more or less spatially extensive [33,34] and composed of weed communities, ruderal vegetation, and deteriorated forested formations. The size of the surveys was determined using descriptive criteria that were in line with the condition of the habitats sampled. Each site was equipped with a 100 m² transect of floristic surveys that ran parallel to the banks of streams and other bodies of water. All the species present for each survey were noted and assigned an abundance-dominance coefficient. The floristic nomenclature used was that of Brunel and Akoegninou [35,36]. Ecological descriptors, such as the overall vegetation cover, the composition and texture of the soil, and the remnants of human activity, were added to the floristic data to supplement them.

*2.4. Data Processing and Analysis*

The acquired floristic data were recorded and adjusted following the national nomenclature. Based on the surveys, a general list of species was produced. Each species' biological and chorological type was recorded, and then the raw and weighted spectrum frequencies were assessed. Afterward, the data were encoded in various matrices in preparation for statistical analysis. The overall floristic balance was established by assessing the specific richness, the spectrum of families, biological forms, and phytogeographical types [37,38].

To perform an indirect analysis of the abiotic ecological factors influencing the distribution of vegetation, the matrix (64 samples plots × 324 species) was subjected to a multivariate principal component test (PCA) using the Statistica 7 software [39]. The floristic data of the four categories of identified wetlands were then subjected to the above analyses.

**3. Results**

*3.1. Overview of Local Floristic Diversity*

Seventy-two (72) plant families were present in total, according to wetlands prospection in the Haho River watershed (Figure 2). This taxonomic procession is dominated by Poaceae (14.95%), followed by Fabaceae (11.98%), Euphorbiaceae (6.20%), Rubiaceae (4.48%), Asteraceae (4.01%), Mimosaceae (3.90%), Cyperaceae (3.85), Arecaceae (3.80%) and Onagraceae (2.76%). Additionally, it should be emphasized that there are families present that are less common but yet typical of moist biotopes, such as Nymphaeaceae (0.93%), Polygonaceae (0.88%), Convolvulaceae (0.83%), Lemnaceae (0.36%), Zingiberaceae (0.26%), Ceratophyllaceae (0.15%), Araliceae (0.10%), and Hydrophyllaceae (0.05%).

From a genus point of view, the floristic diversity of the wetlands of this basin comprises two hundred and thirty-six (236) genera, the most frequent of which are, among others, Ludwigia (3.12%), Panicum (2.60%), Cyperus (2.44%), Elaeis (2.44%), Acacia (2.34%), Paspalum (2.03%), Commelina (1.71%), and Lonchocarpus (1.71%).

The species richness is estimated at three hundred and twenty-three (323) species. The most frequent species seen in this flora are *E. guineensis* Jacq. (2.44%), *P. maximum* Jacq (2.29%), *L. sericeus* (Poir) H. B. K. (1.71%), *Azadirachta indica* A. Juss (1.61%), *Flueggea virosa* Roxb, ex Willd (1.45%), *Millettia thonningii* Schumach. And Thonn (1.45%), *Acacia polyacantha* Willd. (1.30%), *Phyllanthus amarus* Schum. and Thonn (1.19%), *Typha domingensis* Pers. (1.19%), and *Spermacoce ruelliae* DC (1.14%).

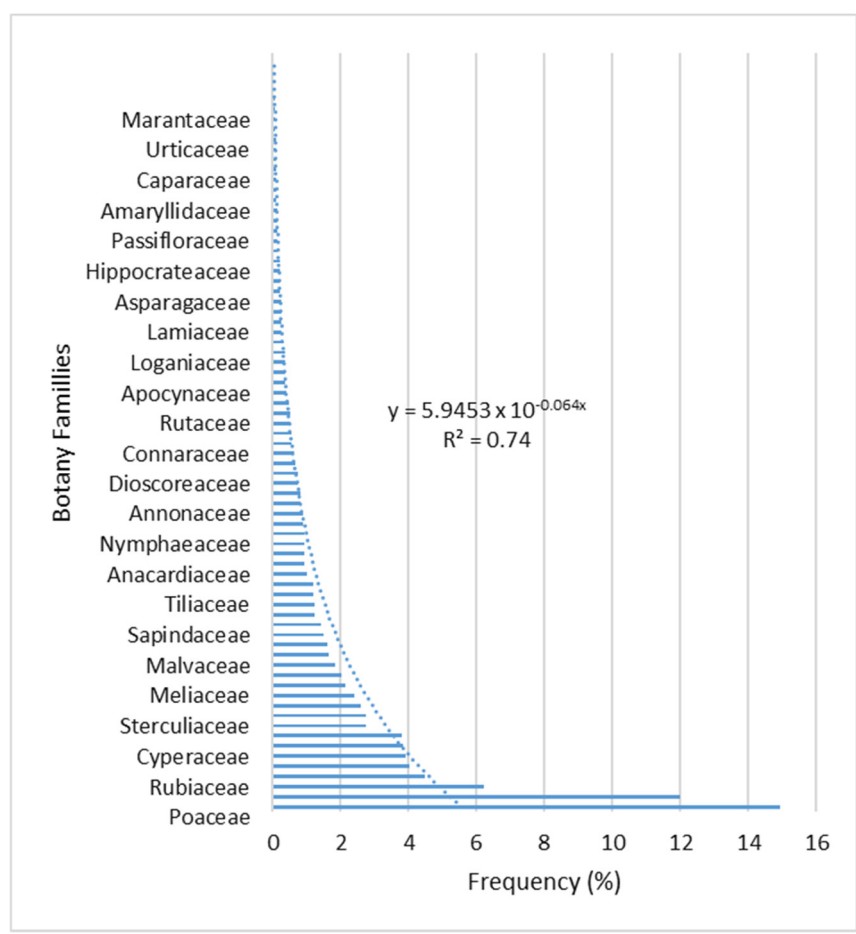

**Figure 2.** Distribution of the relative frequencies of the families identified in the wetlands.

Based on occurrence, micro-phanerophytes (34.70%), followed by therophytes (23.50%), and nano-phanerophytes (10.78%), represent the most frequent biological forms, justifying the character of savannah mosaics that prevail around these wetlands (Figure 3). The weighted spectrum of life forms shows that micro-phanerophytes (38.76%), followed by hemicryptophytes (33.51%), hydrophytes (12.40%), and therophytes (9.45%), dominate and abound in these wetland ecosystems (Figure 3).

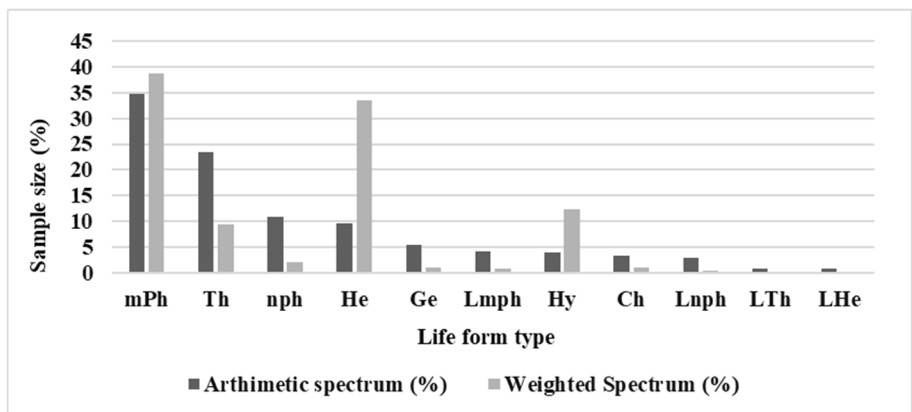

**Figure 3.** Frequency of distribution of species richness by biological types. mPh: micro-phanerophytes. Th: therophytes. nph: nano-phanerophytes. He: hemicrypto-phytes. Ge: geophytes. Lmph: liana-micro-phanerophytes. Hy: hydrophytes. Ch: chamaephytes. Lnph: nano-phanerophyte creepers. LTh: therophyte creepers. LHe: hemicryptophyte.

The raw spectrum of phytochoria (Figure 4) indicates a strong presence of pantropical (31.05%), Guineo-Congolese (21.46%), Sudano-Zambezian (13.86%), and Afro-tropical (10.99%) species. In terms of cover and abundance, Guineo-Congolese (56.97%) and pantropical (20.55%) species almost dominate the flora of these wetlands (Figure 4).

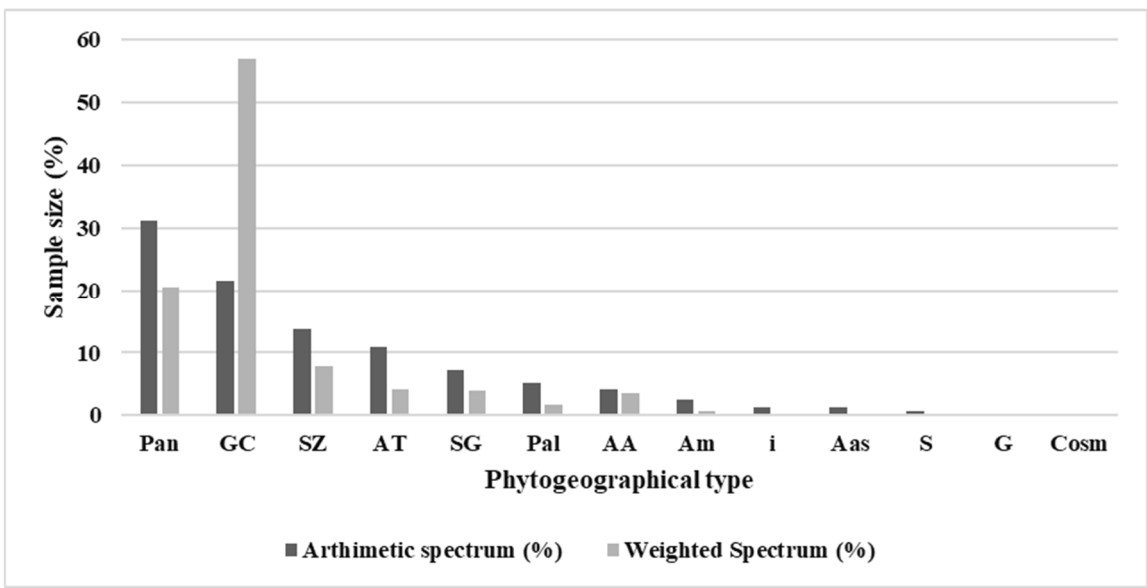

**Figure 4.** Frequency of distribution of species richness by phytogeographical types. Pan: pantropical. GC: Guinean-Congolese species. SZ: Sudano-Zambezian. AT: Afro-tropical. SG: Sudano-Guinean. Pal: paleo-tropical. AA: African-American. Am: Afro-Malagasy. i: introduced. Aas: Afro-Asian. S: Sudanian. G: Guinean. Cosm: Cosmopolitan.

The matrix (324 species × 64 samples plots) was projected onto principal components (Figure 5). It illustrates the distribution of species richness in the composite plane of axes 1 and 2, for which the eigenvalues were significant. Two main gradients influence the distribution of species in the wetland landscape. The first is concerned with the level of immersion indicated by axis 1, which depicts a diminishing gradient of species immersion from right to left, with species on the far right immersed or floating and those on the far left temporarily submerged, depending on the season. A gradient of diminishing edaphic humidity is represented on axis 2. In contrast to stations with negative values, those with positive values have higher relative humidity levels. In other words, the water factor significantly affects how well a species adapts and survives.

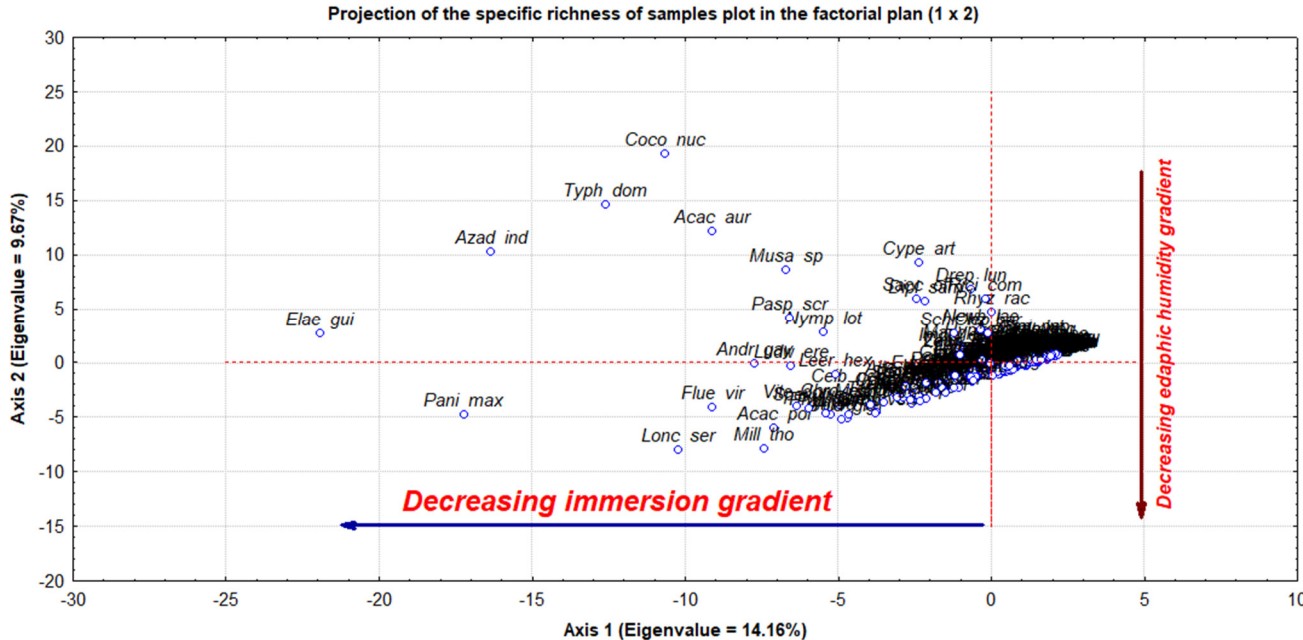

**Figure 5.** Projection of the 324 species using a principal component factorial design based on immersion and relative humidity gradients.

### 3.2. Biological Properties of Main Wetlands Types

#### 3.2.1. Riparian Forests

Riparian wetlands are made up of riparian vegetation on the banks of watercourses and marshy vegetation adjacent to riparian vegetation. Riparian forests were found to harbor two hundred and ninety (290) species, the most common of which are *E. guineensis* Jacq. (2.63%), *P. maximum* Jacq. (2.32%), *L. sericeus* (Poir.) Kunth ex DC. (1.85%), *Flueggea virosa* (Roxb. ex Willd.) Royle (1.70%), *M. thonningii* (Schumach. & Thonn.) Baker (1.70%) and *Paulinia pinnata* L. (1.47%). The floristic procession is divided into two hundred and nineteen (219) genera and sixty-six (66) families. The most frequent families are Poaceae (13.85%), followed by Fabaceae (11.68%), Euphorbiaceae (6.96%), Mimosaceae (4.48%), and Cyperaceae (3.86%).

The raw spectrum of the most prevalent life forms (Figure 6a) was determined to comprise micro-phanerophytes (37.46%), therophytes (21.13%), and nano-phanerophytes (10.52%). As for the most abundant species, the weighted spectrum of life forms shows the dominance of micro-phanerophytes (44.02%), nano-phanerophytes (35.65%), and therophytes (8.13%).

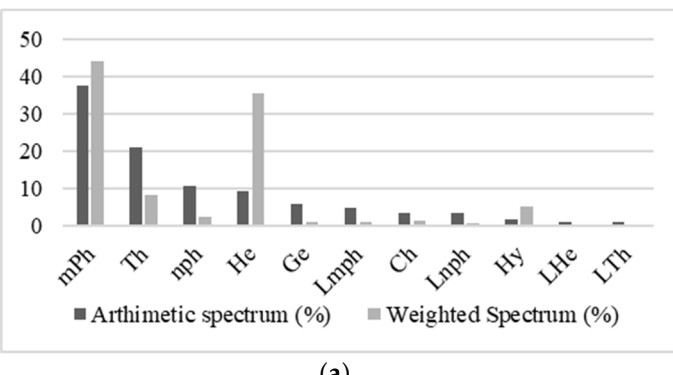

(**a**)

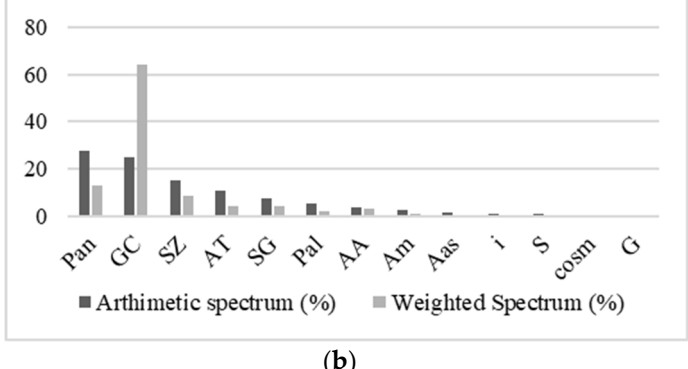

(**b**)

**Figure 6.** Distribution of biological and chorological types in riparian forests: (**a**) LFT: life forms types: mPh: micro-phanerophytes. Th: therophytes. nph: nano-phanerophytes. He: hemicryptophytes. Ge: geophytes. Lmph: liana-micro-phanerophytes. Hy: hydrophytes. Ch: chamephytes. Lnph: lianes nano-

phanerophytes. LTh: liana therophytes. LHe: liana hemicryptophytes. (**b**) TP: phytogeographical types: Pan: pantropicals. GC: Guineo-Congolese species. SZ: Soudano-Zambezian species. AT: Afro-tropicales. SG: Soudano-Guinean species. Pal: paleo-tropicals. AA: Afro-American. Am: Afro-Malgash. i: introduced. Aas: Afro-Asiatics. S: Soudanian, G: Guinean, cosm: cosmopolitan.

The biogeography in this type of wetland shows a high prevalence of raw spectra (Figure 6b) from pantropical (60.91%), Guineo-Congolese (24.84%), and Sudano-Zambezian (15.01%) species. The weighted spectrum is dominated by Guineo-Congolese species (64.21%), followed by pantropical (13.04%) and Sudano-Zambezian (8.60%).

Riparian forests often exhibit three physiognomic units that are significantly connected to the drainage gradient. The first consists of a mosaic lawn of herbaceous plants and grasses with *Ludwigia* sp., *Echinochloa* sp., *Paspalum* sp., and *Cyperus* sp., which grows on the inner banks of rivers close to the flow. The second, which is highly disturbed, is distinguished by an abundance of lianas (*Cissus* sp., *Ampelocissus* sp., *Dioscorea* sp.) in the ligneous formations, with *Pterocarpus santalinoides* L'Hér. ex DC., *Cynometra megalophyla* Harms, *Cola laurifolia* Mast., *Bambusa vulgaris* Schrad. ex J.C.Wendl. in the riparian forests. Finally, the third unit consists of highly disturbed habitats (fallow, savannah, parks) with *E. guineensis* Jacq., *L. sericeus* (Poir.) Kunth ex DC., *P. erinaceus* Poir., *V. paradoxa* C. F. Gaertn., and *M. thonningii* (Schumach. & Thonn.) Baker when the land is relatively drained or of marshy vegetation with *M. inermis* (Willd.) Kuntze, *Polygonum* sp., *Ipomea* sp., *Merinia* sp. and *Typha domingensis* (Pers.) Steud provides a natural receptacle for river overflow during floods. It is associated with *Acacia* sp, *Entada* sp, *Terminalia* sp, and *Combretum* sp on the periphery.

### 3.2.2. The Ponds

The wetland habitats formed by the ponds are also a sanctuary of diversity, made up of physiognomic facies linked to hydrological and edaphic gradients. The majority of sampled ponds have been historically exploited by residents, revealing a low floristic richness and a sizeable presence of plant species with anthropogenic preference.

The specific richness of the ponds consisted of 26 species divided into 26 genera and 18 families. The most representative families were Arecaceae (16.66), followed by Poaceae (16.66) and Cyperaceae (10.41).

The species of *Cocos nucifera* L. (12.50%), *Cyperus articulatus* L. (10.41%), *T. domingensis*, (Pers.) Steud, *A. indica* A. Juss., *Drepanocarpus lunantus* (L.) G. Mey. and *P. scrobiculatum* L. (6.25% each) were the most frequent.

Phytogeographically, the ponds were dominated by pantropical species (46.95%) and Guineo-Congolese (32.44%) (Figure 7b). The most frequent phytochoria were pantropical species (47.91%), followed by Afro-American (12.50%) and introduced species (10.41%). The biological forms frequently encountered were micro-phanerophytes (35.41%), nano-phanerophytes (25%), and hydrophytes (18.75%) (Figure 7b); however, the dominant biological types around the ponds were made up of hydrophytes (37.84%), micro-phanerophytes (35.94%), and hemicryptophytes (20.07%).

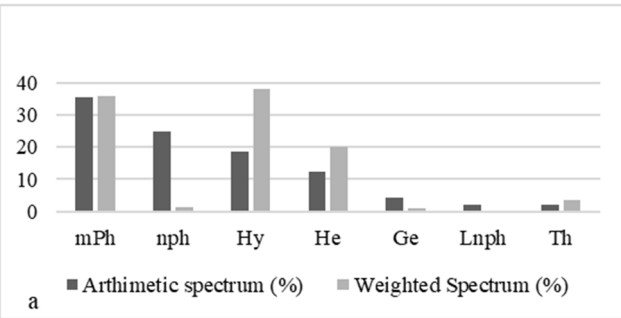
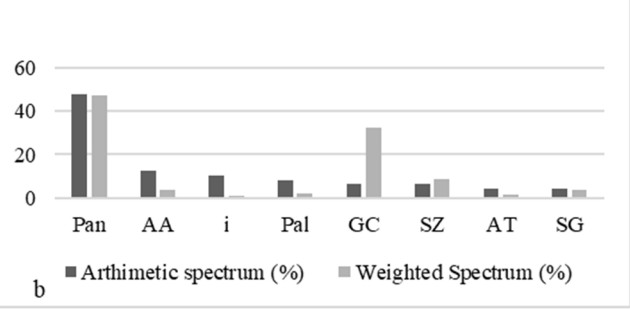

**Figure 7.** Distribution of biological types and phytogeographical types in ponds: (**a**) LFT: life forms types: mPh: micro-phanero0phytes. Th: therophytes. nph: nano-phanerophytes. He: hemicrypto-

phytes. Ge: geophytes. Hy: hydrophytes. Lnph: lianes nano-phanerophytes. (**b**) TP: phytogeographical types: Pan: pantropicals. GC: Guineo-Congolese species. SZ: Soudano-Zambezian species. AT: Afro-tropicales. SG: Soudano-Guinean species. Pal: paleo-tropicals. AA: Afro-American. i: introduced. S: Soudanian. G: Guinean.

While water lilies can be seen in some spots, the ponds are mostly covered with grass, *Typha* sp., in immersion. The pond borders are generally occupied by *Polygonum* sp., *Saccharum* sp. and *Ludwigia* sp. The banks are planted with *A. auriculiformis* A.Cunn. ex Benth., *C. nucifera* L., *E. guineensis* Jacq. and *M. esculenta* Crantz, controlling maintained crops or fallows when they are in operation.

### 3.2.3. Dams and Reservoirs

Reservoirs and dams are artificial wetlands created by mining phosphates and gravel and building dykes (dams) along the banks of rivers and streams. The main purpose of artificial dams and ponds is to produce agricultural goods.

The artificial reservoirs and dams have a specific richness estimated at 184 species divided between 50 families and 147 genera. The most diverse families are Poaceae (17.89%), Fabaceae (13.55%), Mimosaceae, Cyperaceae, and Asteraceae (5.46% each). However, the most frequent species are *P. maximum* Jacq (2.25%), *A. indica* A. Juss., *L. sericeus* (Poir.) Kunth ex DC., *L. erecta* (L.) H. Hara, *T. domingensis* (Pers.) Steud, (1.69% each) and *A. polyacantha.* (1.50%).

The biological forms most frequently encountered are composed of therophytes (33.33%), micro-phanerophytes (27.11%), and hemicryptophytes (10.73%) (Figure 8a); however, the dominant biological types around these artificial water points are made up of hemicryptophytes (34.46%), micro-phanerophytes (28.29%), and hydrophytes (17.78%). Phytogeographically (Figure 8b), wetlands resulting from human activities are dominated by Guineo-Congolese (48.41%) and pantropical (26.52%) species; however, the most frequent phytochoria are pantropical species (36.34%), followed by Guineo-Congolese (15.81%) and Sudano-Zambezian (12.99%).

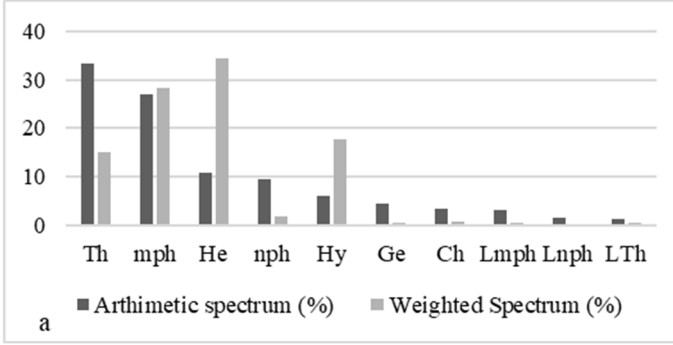
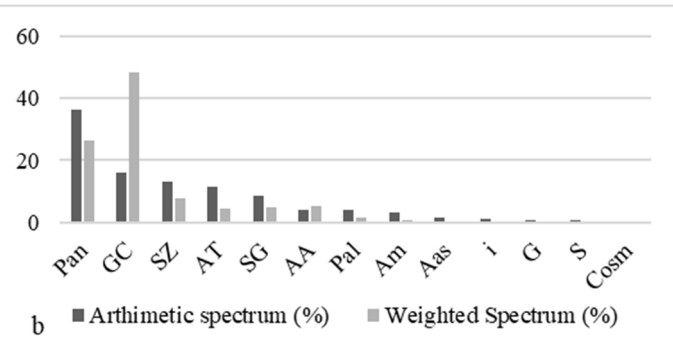

**Figure 8.** Distribution of biological and chorological types of reservoirs (**a**) LFT: life forms types: mph: micro-phanerophytes. Th: therophytes. nph: nano-phanerophytes. He: hemicryptophytes. Ge: geophytes. Lmph: liana-micro-phanerophytes. Hy: hydrophytes. Ch: chamephytes. Lnph: lianes nano-phanerophytes. LTh: liana therophytes. (**b**) TP: phytogeographical types: Pan: pantropicals. GC: Guineo-Congolese species. SZ: Sudano-Zambezian species. AT: Afro-tropicales. SG: Sudano-Guinean species. Pal: paleo-tropicals. AA: Afro-American. Am: Afro-Malgash. i: introduced. Aas: Afro-Asiatics. S: Sudanese. G: Guinean. Cos: cosmopolitan.

Following the increasing drainage gradient, we found on the banks a tree stratum with *Acacia* sp., *Entada* sp. and *P. thonningii* (Schumach) Milne. Redh. The intense runoff gives rise to *Ludwigia* sp., *Paspalum* sp., *Spermacoce* sp., *Cyperus* sp. and *Echinochloa* sp. Water lilies have taken over the reservoirs. *P. lanigerum* R. Br. grasses penetrate undeveloped wetlands, then *T. domingensis* Pers. grasses take over the water's surface.

### 3.2.4. Lakes and Lagoons

The lagoon and lake complex wetlands are closely associated with the ponds in the south of the basin. These types of wetlands are fed by the Boko, Haho, and Zio rivers in connection with the marine system. The floristic report reveals the presence of 21 species divided into 21 genera and 13 families. The most frequent species are *C. nucifera* (12.50%), *T. domingensis* Pers. (10.41%), *E. guineensis* Jacq, *R. racemosa* and *C. articulatus* L. (6.25% each), Arecaceae (18.75%), Mimosaceae, Poaceae and Typhaceae (10.41% each).

Phytogeographically lakes and lagoons are dominated by pantropical species (46.82%) and Guineo-Congolese (26.70%). However, the most frequent phytochoria are pantropical (50%) species, followed by Afro-tropical species (16.66%) and African-American species (10.41%) (Figure 9b). The biological forms frequently encountered are composed of micro-phanerophytes (43.75%), hydrophytes (22.91%), and nano-phanerophytes (18.75%) (Figure 9a); however, the dominant biological types around lakes and lagoons are made up of hydrophytes (40.82%), micro-phanerophytes (25.48%), and hemicryptophytes (10.44%).

Where the bodies of water are brackish, the banks are occupied by degraded mangroves with *Rhizophora* sp., and lawns of *Saccharum* sp., *Paspalum* sp., and *C. articulatus* L. *A. aureum* vegetation is also noticeable in the brackish landscape. In locations with moderate or mild salinity, meadows of *T. domingensis* cover the water bodies, and the banks are colonized by a lawn of *Polygonum* sp., *Saccharum* sp., *Paspalum* sp., and *C. articulatus* L. On hydromorphic soil, the tree stratum is characterized by a strong dominance of coconut groves, palm groves, and reforestation (*A. auriculiformis* A. Cunn. ex Benth).

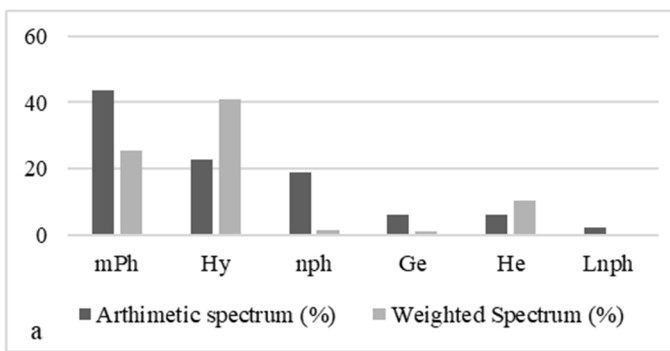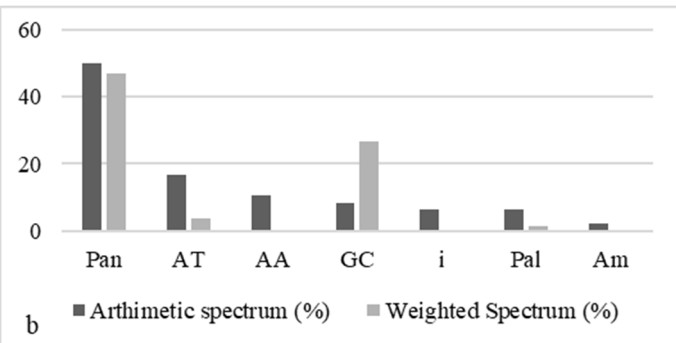

**Figure 9.** Distribution of biological types and chorological types of lakes and lagoons. (**a**) LFT: life forms types: mPh: micro-phanerophytes. nph: nano-phanerophytes. He: hemicryptophytes. Ge: geophytes. Hy: hydrophytes. Lnph: lianes nano-phanerophytes. (**b**) TP: phytogeographical types: Pan: pantropicals. GC: Guineo-Congolese species. AT: Afro-tropicales. Pal: paleo-tropicals. AA: Afro-American. Am: Afro-Malgash. i: introduced.

## 4. Discussion

With 72 families, 236 genera, and 324 species, the wetlands in the Haho River basin have a high floristic potential. Its specific richness exceeds that of the mangroves and associated vegetation of the lagoon–lacustrine complex of the littoral in Togo (23 species) , the swamp forests of Lokoli in Zogbodomey (241 species) in southern Benin [40], and the riparian forests of Burili ulcer (216 species) with a high rate of endemicity in Côte d'Ivoire [41]. Therefore, according to the third criterion of the Ramsar wetland classification related to wetland biodiversity, the study area can be considered a wetland of international importance. This criterion stipulates that: a wetland may be considered of international importance if it supports populations of plant and/or animal species critical to maintaining the diversity of a biogeographic region [42,43]. A further article on the classification of this wetland may be necessary.

Our findings demonstrate that Poaceae and Fabaceae are the dominant plant families in the southern wetlands, whether in Benin or Côte d'Ivoire. Furthermore, the southern

wetlands of these three West African nations share *Ludwigia, Panicum,* and *Cyperus* species. The research areas' shared latitudinal positions and similar climatic regimes are unquestionably responsible for their substantial taxonomic similarities (families and genera) based on occurrence. However, part of the discrepancy in species richness is probably explained by the typological diversity of the wetlands prospected in our study, as opposed to those undertaken in the two other sub-regions [44].

According to the results of several phytosociological studies conducted in the sub-region, biological types are one of the criteria used to evaluate the physiognomy of plant formations [45,46]. The results of this study reveal that micro-phanerophytes and therophytes are the predominant biological forms; this pattern was also seen in the riparian forests in the southern part of Côte d'Ivoire [47]. The strong representation of microphanerophytes would suggest the presence of shrubby formations or open forests close to the sampled locations [33,48]. This is particularly evident in riparian forests and ponds, where woody formations contain forest species such as *P. santalinoides* L'Hér. ex DC., *C. megalophyla* Harms, *Cola laurifolia* Mast., *Bambusa vulgaris* Schrader ex Wendl., *L. sericeus* (Poir.) H. B. K., and *M. thonningii* (Schumach. Et Thonn.) Baker and *M. inermis* (Willd.) Kuntze.

When it comes to therophytes, this preponderance highlights a period of severe vegetation degradation brought on by a sizeable anthropogenic impact on ecosystems [49]. This therophyzation of the floristic richness may indicate the extent pastoralism is expanding in and around these wetlands [50]. The permanence of water, and the expanses of open meadows, explain the growing number of camps of transhumant herders in the catchment area. The abundance of hemicryptophytes around several wetlands in the watershed indicates stable soils and suitable humidity levels, which justify the off-season agriculture that has been recorded there [51,52].

The pantropical and Guineo-Congolese type phytochoria determine the phytogeographical landscape of the study area. The wetlands under investigation exhibit damaged ecosystems and contain signs of human activity. Therefore, ongoing human interference with the ecological dynamism of wetlands has led to the domination of widely dispersed chorological types (pantropical) [53,54]. Additionally, this illustrates how tropical species of the widespread range have enriched the endemic flora of this phytogeographical region (pantropical, Afro-tropical, paleo-tropical). The importance of the Guinean-Congolese groups confirms that the wetlands sampled are at the transition between the Guinean and Congolese climatic domains. However, species having Sudanese characteristics would be found more toward the Sudanese climate region in terms of latitude.

Changes in the characteristics of the floristic composition of wetland ecosystems result from several mutations connected to the forms of use and production [55]. It is acknowledged that wetlands possess a functional dynamic on which the viability of local economies depend (Appendix A). It has also been established that this resilience has spatial and temporal boundaries, where seasonal climate fluctuations also affect the ecological functions of the microorganisms in these particular biotopes by changing the dynamics of wetland formation. In order to stop the disturbance of the food chain in response to the trend of diminishing quality in the services and goods given by wetlands, many studies encourage in-depth research on the ecotoxicology and biological contamination of these biotopes [56].

## 5. Conclusions

This study on the floristic diversity of the wetlands in the Haho River basin found 324 species distributed among 236 genera and 72 families, indicating that this region has high floristic potential. This region could be included in the list of wetland areas of international importance upon confirmation of additional studies in accordance with the Ramsar criteria. The species found belong to the widely dispersed tropical flora (pantropical, Afro-tropical, and paleo-tropical) found to have been substantially increased by the endogenous species. Several factors, including water availability during the season and anthropogenic influences through agriculture, influence the distribution of these species. It

should be noted that the potential of wetlands is undervalued, mainly because the actors are poorly organized and have limited technical and financial resources when implementing the integrated development projects from which these floodable ecosystems can benefit. If wetlands are to be managed sustainably, short- and medium-term actions are required to inform the public authorities of the need for regulation and usage regulation. An integrated management strategy requires action research to categorize wetlands by function (water resource, biodiversity, hunting, or tourism development).

**Author Contributions:** Conceptualization, F.F. and K.W.; methodology, F.F., M.K.; K.F.; resources, M.K.; data curation, K.B., K.W.; writing—original draft preparation, F.F.; writing—review and editing, E.B., K.F.; visualization, F.F.; supervision, K.W., K.B., K.A.; project administration, K.A. All authors have read and agreed to the published version of the manuscript.

**Funding:** This research received no external funding.

**Institutional Review Board Statement:** Not applicable.

**Informed Consent Statement:** Not applicable.

**Data Availability Statement:** Data are available upon request.

**Acknowledgments:** We are very grateful to the Botany and Plant Ecology Laboratory members who contributed to the data collection. TWAS (The World Academy of Sciences), Matsumae International Foundation (MIF), and The Ecosystems Dynamic and Productions (Kyoto University) deserve our gratitude for their hospitality and financial support. Finally, we are indebted to the reviewers of our text, who remain anonymous, for their advice.

**Conflicts of Interest:** The authors declare no conflict of interest.

## Appendix A. Plant Diversity Follow Life Form, Phytogeographical Type and It Endogenous Uses

### LFT: life forms types

mPh: micro-phanerophytes. Th: therophytes. nph: nano-phanerophytes. He: hemicryptophytes. Ge: geophytes. Lmph: liana-micro-phanerophytes. Hy: hydrophytes. Ch: chamaephytes. Lnph: nano-phanerophyte lianas. LTh: lianas therophytes. LHe: hemicryptophyte lianas.

### TP: phytogeographical types

Pan: pantropical. GC: Guinean-Congolese species. SZ: Sudano-Zambezian. AT: Afrotropical. SG: Sudano-Guinean. Pal: paleo-tropical. AA: African-American. Am: Afro-Malagasy. i: introduced. Aas: Afro-Asian. S: Sudanian. G: Guinean. Cos: cosmopolitan.

| No | Species | LFT | TP | EB | L-L | M | R |
|----|---------|-----|-----|-----|-----|---|---|
| 1 | *Abutilon mauritianum* (Jacq) Medik. *** | Th | AT | 1 | | | 1 |
| 2 | *Acacia auriculiformis* A. Cunn. ex Benth. **** ***** | mPh | AT | 1 | 1 | 1 | 1 |
| 3 | *Acacia dudgeoni* Craib ex Hall. ** *** **** ***** | mPh | SZ | | | | 1 |
| 4 | *Acacia polyacantha* Willd. campylacantha (Hochst. ex A. Rich.) Brenan ** *** **** ***** | mPh | SZ | 1 | | | 1 |
| 5 | *Acacia sieberiana* DC.var.villosa ** *** **** ***** | mPh | SZ | 1 | | | 1 |
| 6 | *Acroceras amplectens* Stapf *** | nph | SG | | | | 1 |
| 7 | *Acroceras zizanioides* (Kunth) Dandy *** | He | Pan | | | | 1 |
| 8 | *Adansonia digitata* L. * ** *** ***** ****** | mPh | SZ | 1 | | | 1 |
| 9 | *Adenia rumicifolia* Engl. &Harms ** *** | Lnph | GC | 1 | | | |
| 10 | *Aframomum sceptrum* (Oliv. & D.Ranb.) K.Schum. ** | Ge | GC | | | | 1 |
| 11 | *Ageratum conyzoides* L. ** | Th | Pan | 1 | | | 1 |
| 12 | *Albizia lebbeck* (L.) Benth. ** *** **** ***** | mPh | Pan | 1 | | | |
| 13 | *Albizia zygia* (DC.) J.F. Macbr. ** *** **** ***** | mPh | GC | 1 | | | 1 |
| 14 | *Alchornea cordifolia* (Schum. & Thonn.) Mull. Arg. ** | mPh | GC | 1 | | | 1 |
| 15 | *Allophylus africanus* P. Beauv. ** *** | mPh | GC | | | | 1 |
| 16 | *Alternanthera nodiflora* R.Br. ** | Ch | Pan | 1 | | | 1 |
| 17 | *Alternanthera sessilis* (L.) R. Br.ex Roth ** | Th | Pan | 1 | | | 1 |

| | | | | | | | |
|---|---|---|---|---|---|---|---|
| 18 | *Ampelocissus grantii* (Baker) Planch. * ** | Lmph | SG | 1 | | | 1 |
| 19 | *Ampelocissus leonensis* (Hook.f.) Planch. * ** | Lmph | GC | | | | 1 |
| 20 | *Ampelocissus pentaphylla* (GuilI. & Perr.) Gilg & Brandt * ** | Lmph | SZ | | | | 1 |
| 21 | *Anchomanes difformis* (BIume) Engl. ** | Ge | GC | 1 | | | 1 |
| 22 | *Andropogon gayanus* Kunth ** *** | He | SG | 1 | | 1 | 1 |
| 23 | *Annona senegalensis* Pers. * ** *** **** ***** | nph | SZ | 1 | | | 1 |
| 24 | *Anogeisus leiocarpa* (DC) Guill. Et Perr * ** *** **** ***** | mPh | SZ | 1 | | | 1 |
| 25 | *Anthocleista djalonensis* A. Chev. ** *** | mPh | GC | | | | 1 |
| 26 | *Antiaris toxicaria* Lesch. ssp. welwitschii (Engl.) C.C.Berg ** *** **** ***** | mPh | GC | 1 | | | 1 |
| 27 | *Antidesma venosum* E. Mey. Ex Tul. ** *** **** ***** | mPh | AT | | | | 1 |
| 28 | *Artocarpu saltilis* (Parkinson) Fosberg *   ** *** **** ***** | mPh | i | 1 | | | |
| 29 | *Asparagus africanus* Lam. ** | nph | SZ | | | | 1 |
| 30 | *Aspilia rudus* Oliv. & Hiern ** | Th | SG | | | | 1 |
| 31 | *Asystasia gangetica* (L.) T. Anders. ** *** **** | nph | Pan | | | | 1 |
| 32 | *Axonopus compressus* (Sw.) P.Beauv. ** | Th | Pan | 1 | | | 1 |
| 33 | *Azadirachta indica* A. Juss. ** *** **** ***** | mPh | Pal | 1 | 1 | 1 | 1 |
| 34 | *Bambusa vulgaris* Schrader ex Wendl. | Ge | SG | | | | 1 |
| 35 | *Berlinia grandiflora* (Vah) Hutch. Et Dalziel ** *** **** ***** | mPh | AT | | | | 1 |
| 36 | *Blighia sapida* Konig * ** *** **** ***** | mPh | Pan | | | | 1 |
| 37 | *Boerhavia erecta* L. ** *** | Th | cosm | | | | 1 |
| 38 | *Borassus aethiopum* Mart. * ** *** **** ***** | mPh | SZ | 1 | | | 1 |
| 39 | *Brachiaria deflexa* (Schumach) C. E. Hubbard ex Robyns | Th | Am | 1 | | | |
| 40 | *Bridelia   ferruginea* Benth ** *** **** ***** | mPh | SZ | 1 | | | 1 |
| 41 | *Caesalpinia benthamiana* (Baill.) Herend. Et Zarucchi | nph | AT | | | | 1 |
| 42 | *Calopogonium mucoïdes* Desv.   ** | LTh | Am | 1 | | | 1 |
| 43 | *Calotropis procera* (Aiton) W. T. Aiton ** *** **** ***** ****** | nph | Pal | 1 | | 1 | |
| 44 | *Capsicum frutescens* L. * ** | nph | Pan | | | | 1 |
| 45 | *Carica papaya* L. ** *** | mPh | AA | | | 1 | 1 |
| 46 | *Carissa edulis* Bahl. * ** | nph | Pal | | | | 1 |
| 47 | *Cassia hirsuta* L. ** | | Pan | | | | 1 |
| 48 | *Ceiba pentandra* (L.) Gaertn. * ** *** **** ***** ****** | mPh | Pan | 1 | | 1 | 1 |
| 49 | *Celosia argentea* L.var.argentea | Th | Pal | | | | 1 |
| 50 | *Celosia trigyna* L. ** | Th | Pal | | | | 1 |
| 51 | *Ceratophyllum demersum* L. ** | Hy | Pan | | | | 1 |
| 52 | *Chamaecrista mimosoides* (L.) Greene ** | Th | Pal | 1 | | | 1 |
| 53 | *Chassalia kolly* (Schumach.) Hepper ** | nph | GC | | | | 1 |
| 54 | *Chloris barbata* Sw. ** *** | Th | Pan | 1 | | | 1 |
| 55 | *Chloris pilosa* Schum. *** | Th | Pan | | | | 1 |
| 56 | *Chromolaena odorata* DC. ** *** | nph | Pan | 1 | | 1 | 1 |
| 57 | *Cisampelos mucronata* A. Rich * ** | Lmph | SZ | | | | 1 |
| 58 | *Cissus petiolata* Hook.f. * ** | Lnph | GC | | | | 1 |
| 59 | *Cissus populnea* Guill. Et Perr * ** | LHe | SZ | | | | 1 |
| 60 | *Clausena anisata* (Willd) Hook. F. ex Benth ** *** **** ***** | mPh | AT | | | | 1 |
| 61 | *Cleome rutidesperma* DC. ** | Th | SG | 1 | | | |
| 62 | *Clitoria falcata* Lam ** | Th | AA | 1 | | | 1 |
| 63 | *Cnestis ferruginea* DC. ** *** **** ***** | mPh | GC | | | | 1 |
| 64 | *Cocos nucifera* L. * ** *** **** ***** | mPh | Pan | 1 | 1 | 1 | 1 |
| 65 | *Cola gigantea* A, Chev. Var. gigantea ** *** **** ***** ****** | mPh | GC | 1 | | | 1 |
| 66 | *Cola laurifolia* Mast. ** *** **** ***** | mPh | GC | | | | 1 |
| 67 | *Cola millenii* K. Schum. ** *** **** ***** | mPh | GC | | | | 1 |
| 68 | *Colocasia esculenta* (L.) Schoot * ** | Ge | AT | | | | 1 |
| 69 | *Combretum adenogonium* Steud. Ex A. Rich. ** *** **** ***** | mPh | SG | | | | 1 |
| 70 | *Combretum collinum* Fresen ** *** **** ***** | mPh | SG | | | | 1 |
| 71 | *Combretum glutinosum* Perr.ex De. ** *** **** ***** | mPh | SZ | | | | 1 |

| 72 | *Combretum hispidum* Law. ** *** **** ***** | mPh | GC | | | | 1 |
|---|---|---|---|---|---|---|---|
| 73 | *Combretum molle* R. Br. ex G. Don ** *** **** ***** | mPh | SZ | 1 | | | 1 |
| 74 | *Commelina bracteosa* Hassk. *** | He | Pan | | | | 1 |
| 75 | *Commelina diffusa* Burm.f. *** | He | Pan | 1 | | | 1 |
| 76 | *Commelina erecta* L. *** | Ch | AT | 1 | | | 1 |
| 77 | *Crinum jagus* (J. Thomps.) Dandy. ** | Ge | GC | 1 | | | 1 |
| 78 | *Crossopteryx febrifuga* (Afzel. ex G. Don) Benth ** *** **** ***** | mPh | SZ | | | | 1 |
| 79 | *Crotalaria cephalotes* Steud. ex A.Rich. *** | Th | Pan | 1 | | | 1 |
| 80 | *Crotalaria retusa* Linn *** | Ch | Pan | 1 | | | 1 |
| 81 | *Croton lobatus* L. *** | Th | Pan | 1 | | | 1 |
| 82 | *Culcasia scandens* P. Beauv. *** | Lmph | GC | | | | 1 |
| 83 | *Cussonia kirkii* Seeman var. kirkii ** *** **** ***** | mPh | SZ | | | | 1 |
| 84 | *Cyanotis lanata* Benth. ** | He | SG | | | | 1 |
| 85 | *Cynodon dactylon* (L.) Pers ** | Th | Pan | 1 | | | |
| 86 | *Cynometra megalophylla* Harms ** *** **** ***** | mPh | GC | | | | 1 |
| 87 | *Cyperus alternifolius* L. *** | Ge | i | 1 | | | 1 |
| 88 | *Cyperus articulatus* L. *** | Hy | i | 1 | 1 | 1 | 1 |
| 89 | *Cyperus distans* L.f. s.l. *** | Ge | Pan | 1 | | | 1 |
| 90 | *Cyperus haspan* Linn *** | Ge | Pan | 1 | | | 1 |
| 91 | *Cyperus latifolius* Poir. *** | Ge | Pan | | | | 1 |
| 92 | *Cyperus rhotondus* Linn. *** | Ge | Pan | 1 | | | 1 |
| 93 | *Cyperus tenuiculmis* Boeck.s.l. *** | He | SG | 1 | | | 1 |
| 94 | *Cyphostemma adenocole* (Steud.) Desc. ** | Lmph | AT | | | | 1 |
| 95 | *Cyphostemma sokodense* (Gilg & Brandt) Desc. ** | Lmph | G | | | | 1 |
| 96 | *Dactyloctenium aegyptium* (L.) Willd. *** | Th | Pan | 1 | | | 1 |
| 97 | *Daniellia oliveri* (Rolfe) Hutch & Dalz ** *** **** ***** ****** | mPh | SG | | | | 1 |
| 98 | *Delonix regia* (Boj. Ex Hook.) Raf.  **** ***** | mPh | Am | | 1 | | |
| 99 | *Desmodium adscendens* (Sw.) DC var. adscendens ** | Th | AA | | | | 1 |
| 100 | *Desmodium barbatum* (L.) Benth. var. dimorphum ** | Th | AT | | | | 1 |
| 101 | *Desmodium salicifolium* (Poir.) DC. ** | nph | GC | 1 | | | |
| 102 | *Desmodium tortuosum* (Sw.) DC. ** | nph | Pan | 1 | | | 1 |
| 103 | *Desmodium triflorum* (L.) DC ** | Lnph | Pan | 1 | | | 1 |
| 104 | *Desmodium velutinum* (Willd.) DC. ** | Th | Pal | | | | 1 |
| 105 | *Dialium guineense* Willd. * ** *** **** ***** ****** | mPh | GC | | | | 1 |
| 106 | *Dicrostachys cynera* (L.) Wight & Arn *** | nph | AT | 1 | 1 | | 1 |
| 107 | *Digitaria ciliaris* (Retz.) Koeler *** | Th | Pan | | | | 1 |
| 108 | *Digitaria horizontalis* Willd. *** | Th | AA | 1 | | | 1 |
| 109 | *Dioscorea bulbifera* L. ** | Ge | Pan | 1 | | | 1 |
| 110 | *Dioscorea togoensis* Knuth ** | Ge | GC | | | | 1 |
| 111 | *Diospyros mespiliformis* Hochst.ex A.DC. * ** *** **** ***** ****** | mPh | SZ | | | | 1 |
| 112 | *Diospyros monbuttensis* Gürke * ** *** **** ***** ****** | mPh | GC | | | | 1 |
| 113 | *Diplazium sammatii* (Kuhn) C. Chr. ** | Hy | AT | 1 | 1 | | 1 |
| 114 | *Dracaena arborea* (Willd.) Link. ** ****** | mPh | GC | | | | 1 |
| 115 | *Drepanocarpus lunantus* ** | nph | AA | | 1 | 1 | |
| 116 | *Drypetes floribunda* (Muell.–Arg.) Hutch. *** *** **** ***** | mPh | GC | | | | 1 |
| 117 | *Echinochloa colona* (L.) Link ** *** | Th | Pan | 1 | | | 1 |
| 118 | *Echinochloa obtusiflora* Stapf ** *** | Th | Pan | 1 | | | |
| 119 | *Echinochloa pyramidalis* (Lam.) Hitchc. & Chase ** *** | Th | Pan | 1 | | | 1 |
| 120 | *Eclipta prostata* L. ** *** | Th | GC | 1 | | | 1 |
| 121 | *Ehretia cymosa* Thonn. Ex Sehum. var. cymosa Brenan ** *** | mPh | GC | | | | 1 |
| 122 | *Eichhornia crassipes* (Mart.) Solms-Laub.** *** | Hy | Pan | 1 | | | 1 |
| 123 | *Elaeis guineensis* Jacq. * ** *** **** ***** | mPh | GC | 1 | 1 | 1 | 1 |
| 124 | *Eleusine indica* Gaertn *** | Th | Pan | 1 | | | 1 |
| 125 | *Emilia coccinea* (Sims.) G. Don *** | Th | SG | 1 | | | 1 |

| 126 | *Entada africana* Guill et Perr ** *** **** ***** | mPh | SZ | 1 | | 1 |
|-----|--------------------------------------------------|-----|----|---|---|---|
| 127 | *Entidesma venosum* E. Mey. Ex Tul. *** | nph | AT | | | 1 |
| 128 | *Eragrostis ciliaris* (Linn.) R. Br.*** | Th | Pan | | | 1 |
| 129 | *Eragrostis tremula* (Lam.) Steud. *** | Th | Pan | 1 | | |
| 130 | *Eriosema griseum* Baker var. griseum | Th | AA | 1 | | 1 |
| 131 | *Euphorbia heterophylla* L. *** | Th | Pan | 1 | | 1 |
| 132 | *Euphorbia hirta* Linn. ** *** | Th | Pan | 1 | | 1 |
| 133 | *Euphorbia hyssopifolia* Linn ** *** | Th | Pan | | | 1 |
| 134 | *Ficus asperlifolia* Miq. ** *** **** ***** | nph | GC | | | 1 |
| 135 | *Ficus dicranostyla* Mildbr. * ** *** **** ***** ****** | mPh | GC | | | 1 |
| 136 | *Ficus elastica* Roxb. ** *** **** ***** | mPh | AT | | | 1 |
| 137 | *Ficus exasperata* Vahl. ** *** **** ***** | mPh | AT | | | 1 |
| 138 | *Ficus lutea* Vahl ** *** **** ***** | mPh | i | | 1 | |
| 139 | *Ficus mucuso* Ficalho ** *** **** ***** | mPh | GC | 1 | | 1 |
| 140 | *Ficus sur* Forssk. * ** *** **** ***** ****** | mPh | SG | 1 | | 1 |
| 141 | *Ficus sycomorus* L. * ** *** **** ***** ****** | mPh | SZ | | | 1 |
| 142 | *Fimbristylis littoralis* Gaudet *** | He | Pan | 1 | | 1 |
| 143 | *Flacourtia indica* (Burm.f.) Merr ** *** **** | nph | Aas | | | 1 |
| 144 | *Fleurya aestuans* (Linn.) ex Miq ** *** | Th | Pan | | | 1 |
| 145 | *Flueggea virosa* (Roxb, ex Willd) Voigt ** | nph | Pan | 1 | 1 | 1 |
| 146 | *Fuirena umbellata* Rottb.** | Ge | Pan | 1 | | 1 |
| 147 | *Gmelina arborea* Roxb ** *** **** ***** | mPh | i | | | 1 |
| 148 | *Gomphrena celosioides* Mart. ** | Ch | cosm | 1 | | 1 |
| 149 | *Grewia venusta* fresen * ** *** **** ***** | mPh | SZ | 1 | | 1 |
| 150 | *Griffonia simplicifolia* (Val. Ex DC) Baill. | Lmph | AT | | | 1 |
| 151 | *Hewittia scandens* (Milne) Mabberley | Lnph | Pan | 1 | | 1 |
| 152 | *Hexalobus monopetalus* (A. Rich.) Engl. & Diels. * ** *** **** ***** | mPh | SZ | | | 1 |
| 153 | *Hibiscus asper* Hook. F. * ** | nph | Pan | 1 | | 1 | 1 |
| 154 | *Hibiscus Ionchosepalus* Hochr. * ** | nph | Pan | 1 | | |
| 155 | *Holarrhena floribunda* (G. Don) Durand et Sching ** *** **** ***** | mPh | AT | | | 1 |
| 156 | *Hoslundia opposita* Bahl. ** | nph | Am | | | 1 |
| 157 | *Hydrolea macrosepala* A.W. Benn. ** | Th | AT | 1 | | |
| 158 | *Hyparrhenia involucrata* Stapf. *** | Th | SG | 1 | | 1 |
| 159 | *Hyppocratea indica* Planch.** | Lnph | Aas | | | 1 |
| 160 | *Hyptis suaveolens* (L.) Poit. ** | Th | Pan | 1 | | |
| 161 | *Imperata cylindrica* var africana (Aderss.) ** *** | He | Pan | 1 | | 1 |
| 162 | *Indigofera astragalirea* ** | Th | Pan | 1 | | |
| 163 | *Indigofera dendroides* Jacq. *** | nph | SZ | | | 1 |
| 164 | *Indigofera hirsuta* L. var. hirsuta ** | Ch | Pan | 1 | | 1 |
| 165 | *Indigofera suffruticosa* Mill. ** | nph | Pan | | | 1 |
| 166 | *Indigofera tinctoria* L. var. tinctoria ** | nph | Pan | 1 | | 1 |
| 167 | *Ipomea aquatica* Forssk ** *** | Lnph | GC | | | 1 |
| 168 | *Ipomea involucrata* (P.) Beouv.** *** | Th | AT | 1 | | |
| 169 | *Ipomea vagans* Bak. ** *** | Th | AT | 1 | | |
| 170 | *Khaya senegalensis* (Desr.) A. Juss. ** *** **** ***** | mPh | SZ | 1 | | 1 |
| 171 | *Kigelia africana* (Lam.) Benth. ** *** **** ***** ****** | mPh | GC | 1 | | 1 |
| 172 | *kyllinga bulbosa* Beauv. ** | Ge | Pan | 1 | | 1 |
| 173 | *Kyllinga pumila* Michx ** | Ge | Pan | 1 | | 1 |
| 174 | *Kyllinga squamulata* Thonn. ex Vahl ** | Ge | Pan | | | 1 |
| 175 | *Lannea acida* A. Rich. * ** *** **** ***** | mPh | GC | | | 1 |
| 176 | *Lannea barteri* (Oliv.) Enhl. * ** *** **** ***** | mPh | GC | 1 | | 1 |
| 177 | *Launaea taraxacifolia* (Willd.) Amin ex C. Jeffrey * ** *** | Th | GC | 1 | | 1 |
| 178 | *Lecaniodiscus cupanioides* Planch. ex Benth. ** *** **** ***** | mPh | GC | | | 1 |
| 179 | *Leersia hexandra* Sw. ** | He | Pan | 1 | | 1 |

| | | | | | | | |
|---|---|---|---|---|---|---|---|
| 180 | *Lemna aequinoctialis* Welw | Hy | Pan | 1 | | | 1 |
| 181 | *Leptadenia hastata* (Pers.) Decine ** *** | Lmph | SZ | 1 | | | 1 |
| 182 | *Leptochloa caerulenscens* Steud. ** *** | Th | SG | 1 | | | 1 |
| 183 | *Leucaena leucocephala* (Lam.) de Wit ** *** **** ***** | mPh | Pan | 1 | 1 | | 1 |
| 184 | *Lippia multiflora* Moldenke ** *** **** ***** | nph | SG | | | | 1 |
| 185 | *Lippia rugosa* A.Chev. ** *** **** ***** | nph | SZ | | | | 1 |
| 186 | *Lonchocarpus sericeus* (Poir.) H. B. K. ** *** **** ***** | mPh | GC | 1 | | | 1 |
| 187 | *Lophira lanceolata* Van Tiegh. Ex Keay ** *** **** ***** | mPh | GC | | | | 1 |
| 188 | *Ludwigia abyssinica* A. Rich ** *** | Th | GC | 1 | | | 1 |
| 189 | *Ludwigia decurrens* Walt. ** *** | Th | AA | 1 | | | 1 |
| 190 | *Ludwigia erecta* (L.) Hara ** *** | Th | AA | 1 | | 1 | 1 |
| 191 | *Ludwigia hyssopifolia* (G. Don.) Exell ** *** | Th | AT | | | | 1 |
| 192 | *Ludwigia octavalvis* (Jacq) P. Raven ** *** | Th | AT | 1 | | | 1 |
| 193 | *Ludwigia stolonifera* (Guill. & Perr.) Raven ** *** | Hy | Aas | 1 | | | 1 |
| 194 | *Luffa cylindrica* (L.) M.J. Roem. ** *** | Th | Pan | 1 | | | 1 |
| 195 | *Mallotus oppositifolius* (Geisel.)Müll.Arg. ** *** | mPh | Am | 1 | | | 1 |
| 196 | *Malvastrum coromandelianum* (L.) Garcke. ** *** | Th | Pan | | | | 1 |
| 197 | *Mangifera indica* Sp. Pl. * ** *** **** ***** | mPh | Aas | 1 | | | 1 |
| 198 | *Manihot esculenta* Crantz * ** *** | nph | Pan | | | 1 | |
| 199 | *Margaritaria discoïdea* (Baill.) Webster ** *** | mPh | AT | 1 | | | 1 |
| 200 | *Mariscus alternifolius* Vahl. ** *** | He | Pan | | | | 1 |
| 201 | *Mayetenus senegalensis* (Lam.) Exell. ** *** **** ***** | mPh | SZ | | | | 1 |
| 202 | *Melanthera scandens* (Schum. & Thonn.) Roberty | Lmph | AT | 1 | | | 1 |
| 203 | *Melochia corchorifolia* L. ** *** | Ch | Pan | 1 | | | 1 |
| 204 | *Melochia melissifolia* Benth. ** *** | nph | Pan | | | | 1 |
| 205 | *Merinia umbellata* (L.) Hallier. F. ** *** | Th | Pan | 1 | | | |
| 206 | *Mezoneuron benthamianum* Baill ** *** | Lmph | GC | 1 | | | 1 |
| 207 | *Milicia excelsa* (Welw.) C. C. Berg ** *** **** ***** ****** | mPh | GC | 1 | | | 1 |
| 208 | *Millettia thonningii* (Schumach. Et Thonn.) Baker ** *** **** ***** | mPh | GC | 1 | | | 1 |
| 209 | *Mimosa pudica* L. ** *** | nph | Pan | | | | 1 |
| 210 | *Mirremia aegyptiaca* (L.) Urban ** *** | Th | Pan | | | | 1 |
| 211 | *Mitragyna inermis* (Willd.) Kuntze ** *** **** ***** ****** | mPh | SZ | 1 | | 1 | 1 |
| 212 | *Mnesithea granularis* (L.) Koning & Sosef ** *** | Th | Pan | | | | 1 |
| 213 | *Momordia charantia* L. ** *** | Lnph | Pan | | | | 1 |
| 214 | *Monechma ciliatum* (Jacq.) Milne-Redh. ** *** | nph | AT | | | | 1 |
| 215 | *Morelia senegalensis* A. Rich. Ex DC. ** *** | nph | SG | | | | 1 |
| 216 | *Morinda* lucida Sp. Pl. ** *** **** ***** | mPh | Pan | 1 | | | 1 |
| 217 | *Musa* sp L. * ** *** | Ge | Pan | 1 | 1 | 1 | 1 |
| 218 | *Newbouldia laevis* (P. Beauv.) Seem ** *** **** ***** ****** | mPh | GC | 1 | | | 1 |
| 219 | *Nymphae lotus* Linn. ** | Hy | AT | 1 | 1 | 1 | 1 |
| 220 | *Nymphea maculata* Schum. & Thonn. ** | Hy | AT | 1 | | | 1 |
| 221 | *Oncoba gilgiana* Sprague Bull. Herb. Boiss. ** *** | nph | GC | | | | 1 |
| 222 | *Oplimenus burmannii* (Retz) P. Beauv. | Th | SZ | 1 | | | 1 |
| 223 | *Oriza barthii* A. Chev. * ** *** | He | SZ | 1 | | 1 | 1 |
| 224 | *Oslandia opposita* ** *** | nph | Am | | | | 1 |
| 225 | *Panicum laxum* SW. ** *** | Th | AA | | | | 1 |
| 226 | *Panicum maximum* Jacq. ** *** | He | GC | 1 | 1 | 1 | 1 |
| 227 | *Panicum tenellum* Lam. ** *** | Th | SZ | 1 | | | |
| 228 | *Parkia biglobosa* (Jacq) R. Br. Ex Benth * ** *** **** ***** | mPh | SZ | 1 | | | 1 |
| 229 | *Paspalum polystachion* (L.) Schult.subsp.polystachion L. ** *** | Th | GC | 1 | | | 1 |
| 230 | *Paspalum scrobiculatum* L. ** *** | He | Pan | 1 | 1 | 1 | 1 |
| 231 | *Paspalum vaginatum* Sw. ** *** | Th | Pan | 1 | | | 1 |
| 232 | *Passiflora foetida* L. ** *** | Lnph | Pan | | | | 1 |
| 233 | *Paulinia pinnata* L. * ** *** | Lmph | AT | 1 | | | 1 |

| 234 | *Pavetta corymbosa* (DC.) F. N. Williams ** *** | mPh | SG | | | | 1 |
|-----|---------------------------------------------------|------|-----|---|---|---|---|
| 235 | *Pennisetum hordoides* (Lam.) Steud  ** *** | Th | GC | 1 | | | 1 |
| 236 | *Pennisetum purpureum* Schumach. ** *** | Th | Pan | | | | 1 |
| 237 | *Pentodon pentandrus* (Schum. & Thonn.) Vatke ** *** | Th | AT | 1 | | | 1 |
| 238 | *Pergularia daemia* (FORSSK.) Chiov. ** *** | Lmph | AT | 1 | | | 1 |
| 239 | *Persicaria senegalensis* (Meisn.) Sojak | nph | AT | 1 | | | |
| 240 | *Phoenix reclinata* Jacq. *** **** | mPh | AT | 1 | | | 1 |
| 241 | *Phyllanthus amarus* Schum. Et Thonn ** | Th | Pan | 1 | | | 1 |
| 242 | *Physali angulata* L. ** *** | Th | Pan | 1 | | | 1 |
| 243 | *Physalis minima* L. ** *** | Th | Pan | 1 | | | |
| 244 | *Piliostigma thonningii* (Schumach) Milne. Redh. ** *** **** ***** | mPh | S | 1 | | | 1 |
| 245 | *Polygonum lanigerum* R. Br. ** *** | nph | Aas | 1 | | | 1 |
| 246 | *Polygonum salicifolium* Brouss. Ex Willd. ** *** | Th | AT | 1 | | | 1 |
| 247 | *Polygonum senegalense* Meisn. ** *** | He | AT | 1 | | | 1 |
| 248 | *Pouteria alnifolia* (Baker) Pierre ** *** **** ***** | mPh | GC | 1 | | | 1 |
| 249 | *Premna quadrifolia* Schumach. Et Thonn ** *** | nph | GC | 1 | | | 1 |
| 250 | *Prosopis africana* (Guill. & Perr.) Taub. ** *** **** ***** | mPh | SZ | | | | 1 |
| 251 | *Pseudocedrela kotschyi* (Schweinf.) Harms ** *** **** ***** | mPh | SZ | 1 | | | 1 |
| 252 | *Pterocarpus erinaceus* Poir. ** *** **** ***** | mPh | GC | 1 | | | 1 |
| 253 | *Pterocarpus santalinoides* L'Hér.ex DC. ** *** **** ***** | mPh | GC | 1 | | | 1 |
| 254 | *Rauvolfia vomitoria* Afzel. ** *** **** ***** | mPh | GC | | | | 1 |
| 255 | *Reissantia indica* (Willd.) N. Hallé var. loeseneriana (Hutch.&M.B.Moss) N.Hallé ** **** | Lnph | Aas | | | | 1 |
| 256 | *Rhyzophora racemosa* G. Mey. ** *** **** ***** | mPh | AA | | 1 | | |
| 257 | *Ricinus communis* L. ** *** | nph | Pan | | 1 | 1 | |
| 258 | *Ritchiea reflexa* (Thonn.) Gilg & Benedict ** *** | nph | Pan | | | | 1 |
| 259 | *Rottboellia cochinchinensis* (Lour.)W.D.Clayton ** *** | Th | Pal | 1 | | | 1 |
| 260 | *Rourea coccinea* (Thom, ex Schumach) ** *** | Lnph | AT | 1 | | | 1 |
| 261 | *Saba comorensis* (Boj.)Pichon ** *** | Lmph | Am | | | | 1 |
| 262 | *Saccharum officinarum* L. * ** *** **** ***** | nph | Pan | 1 | 1 | 1 | |
| 263 | *Sacciolepis africana* C. E. Hubb. Et Snowden | He | SZ | | | | 1 |
| 264 | *Samanea saman* (Jacq.) Merr. *** *** **** ***** | mPh | AA | | | | 1 |
| 265 | *Sarcocephalus latifolus* (sm) E. A. Bruce * ** *** **** ***** | mPh | AT | 1 | | | 1 |
| 266 | *Schrankia leptocarpa* DC. ** *** | Lnph | Pan | 1 | 1 | 1 | 1 |
| 267 | *Scoparia dulcis* (L.) ** *** | Th | SZ | 1 | | | |
| 268 | *Secamone afzelii* (Schult.) K. Schum. ** *** | Lmph | GC | 1 | | | 1 |
| 269 | *Securidaca longepedunculata* Fresen. ** *** **** ***** | mPh | SZ | | | | 1 |
| 270 | *Sena siamea* (Lam.) H. S. Irwin et Barneby ** *** **** ***** | mPh | Pan | 1 | | | 1 |
| 271 | *Sesbania dalzielii* Phillip. & Hutch ** *** | Th | Pan | 1 | | | |
| 272 | *Sesbania sesban* (L) Merr. Var. punctata (DC) J.B. Gillett ** *** | Th | GC | | | | 1 |
| 273 | *Setaria barbata* (Lam) Kunth ** *** | Th | Pal | 1 | | | 1 |
| 274 | *Sida acuta* Burm. F. ** *** | nph | Pan | 1 | | | 1 |
| 275 | *Sida corymbosa* R. E. Fries ** *** | Th | GC | | | | 1 |
| 276 | *Sida linifolia* Juss ex Cav. ** *** | Th | AA | | | | 1 |
| 277 | *Sida rhombifolia* L. ** *** | nph | GC | | | | 1 |
| 278 | *Solenostemon monostachyus* (P.Beauv.) Briq. Ssp monostachyus ** *** | Th | GC | | | | 1 |
| 279 | *Sorghum arundinaceum* (Willd.)Stapf * ** *** | He | SG | 1 | | | 1 |
| 280 | *Spathodea campanulata* P.Beauv. ** *** **** ***** | mPh | GC | | | | 1 |
| 281 | *Spermacoce ruelliae* DC. ** *** | Th | SZ | 1 | | | 1 |
| 282 | *Sphenoclea zeylanica* Gaertn. ** *** | Hy | Pan | 1 | | | |
| 283 | *Spigelia anthelmia* L. ** *** | Th | AA | 1 | | | 1 |
| 284 | *Spondias cytherea* Sonner. ** *** **** ***** | mPh | Pan | | | | 1 |
| 285 | *Spondias mombin* L. * ** *** **** ***** | mPh | Pan | | | | 1 |
| 286 | *Sporobolus pyramidalis* P. beauv ** *** | He | SZ | 1 | | | 1 |
| 287 | *Stachytarpheta indica* (Linn) Vahl. ** *** | Th | Am | 1 | | | 1 |

| 288 | *Sterculia setigera* Delile. * ** *** **** ***** ****** | mPh | SZ | 1 | | | |
|---|---|---|---|---|---|---|---|
| 289 | *Sterculia tragacantha* Lindl. ** *** **** ***** | mPh | GC | | | | 1 |
| 290 | *Sterospermum kunthianum* Cham. ** *** **** ***** | mPh | SG | 1 | | | 1 |
| 291 | *Stylochaeton hypogaeum* Lepr. ** *** | Ge | SZ | 1 | | | 1 |
| 292 | *Synedrella nodiflora* (L.) Gaertn. ** *** | Th | Pan | 1 | | | 1 |
| 293 | *Tacca leontopetaloides* (L.) O. Ktze. ** *** | Ge | Pal | | | | 1 |
| 294 | *Talinum triangulare* (Jacq.) Willd. * ** *** | Ch | Pal | | | | 1 |
| 295 | *Tectona grandis* L. F. ** *** **** ***** | mPh | Pal | | | | 1 |
| 296 | *Tephrosia bracteolata* Guill. & Perr. | nph | SG | 1 | | | 1 |
| 297 | *Tephrosia elegans* Schumach. ** *** | nph | SG | 1 | | | |
| 298 | *Tephrosia platycarpa* Guill.&Perr. ** *** | nph | SG | 1 | | | |
| 299 | *Tephrosia villosa* (L.) Pers. ** *** | nph | Pan | 1 | | | |
| 300 | *Terminalia glaucescens* Planch.ex Benth. ** *** **** ***** | mPh | SG | 1 | | | 1 |
| 301 | *Terminalia mollis* Laws. ** *** **** ***** | mPh | SZ | | | | 1 |
| 302 | *Tiliacora funifera* (Miers) OIiv. ** *** | Lmph | GC | | | | 1 |
| 303 | *Trachyphrynium braunianum* (K.Schum.) Bak. | Lmph | GC | | | | 1 |
| 304 | *Trema orientalis* (L.) Blume. ** *** **** ***** | mPh | GC | | | | 1 |
| 305 | *Trianthema portulacastrum* L. ** *** | Th | Pan | 1 | | | |
| 306 | *Triclisia subcordata* Oliv. ** *** | Lnph | GC | | | | 1 |
| 307 | *Tridax procumbens* L. ** *** | Th | Pan | 1 | | | 1 |
| 308 | *Triumfetta rhomboidea* Jacq. ** *** | nph | Pan | | | | 1 |
| 309 | *Typha domingensis* Pers. ** *** | Hy | Pan | 1 | 1 | 1 | 1 |
| 310 | *Uraria picta* (Jacq.) DC. ** *** | nph | Pal | 1 | | | 1 |
| 311 | *Urena lobata* L. ** *** | nph | G | 1 | | | 1 |
| 312 | *Uvaria chamae* P. Beauv. ** *** | Lmph | GC | | | | 1 |
| 313 | *Vernonia amidalina* Del. * ** *** | nph | GC | | | | 1 |
| 314 | *Vernonia cinerea* (L.) Less. * ** *** | Th | Pan | 1 | | | 1 |
| 315 | *Vernonia colorata* (Willd) Drake. * ** *** | nph | SZ | 1 | | 1 | 1 |
| 316 | *Vigna radiata* (L.)R.Wilczek var.radiata | Th | Pan | 1 | | | 1 |
| 317 | *Vigna trichocarpa* (C.Wright) A.Delgado ** *** | Th | Pan | | | | 1 |
| 318 | *Vitellaria paradoxa* C.F.Gaertner subsp. Paradoxa * ** *** **** ***** | mPh | SG | | | | 1 |
| 319 | *Vitex doniana* Sweet. * ** *** **** ***** | mPh | SZ | 1 | | | 1 |
| 320 | *Waltheria indica* L. ** *** | nph | Pan | 1 | | | 1 |
| 321 | *Xanthosoma mafaffa* Sehott * ** *** | Ge | Pan | | | | 1 |
| 322 | *Zanha golungensis* Hiern. ** *** **** ***** | mPh | SG | | | | 1 |
| 323 | *Zanthoxylum Zanthoxyloides* (Lam.) Zepernick et Timler * ** *** **** ***** | mPh | SG | 1 | | 1 | 1 |

EB: ponds and dams. L-L: lakes and lagoons. M: ponds. R: riparian forests. Endogenous uses: * (food), ** (medecine), *** (forrage), **** (fuel wood), ***** (agroforestry use), ****** (magio-religious use).

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
