# Peer review of "Flora and Typology of Wetlands of Haho River Watershed, Togo"

_sustainability, doi:10.3390/su15032814_

Round 1

Reviewer 1 Report

I find this work to be quite sound as far as it goes, but I also think it should go further for Sustainability.  The paper is well written, the methods used are standard and uncontroversial and the statistical results are sound as far as I can infer.  The tables and figures are also of good quality.  As written, however, the paper is a major vegetation analysis using standard techniques, and doesn't make for a strong nexus with Sustainability.  

That is, of all the species of plants recorded, which ones do local people use and for what purposes?  Are there extractive rules in place? The authors mention the large population of people in the region and that wetlands are important for providing ecosystem services, but they don't show up in this work subsequently.  

I recommend they add a bit of text by, for example, pointing out the uses of various plants for textiles, foods, making baskets, mats, fencing/walling materials, etc in the Appendix, and putting some of that verbiage in the Intro, Results and Discussion (just one or two sentences each) and then amending the Appendix.  I would furthermore encourage the authors to prepare an additional follow-up paper that just deals with those examples of the floristics of wetlands in the Togo-Benin region that are economically important at local or national scales, and address whether the consumptive uses are, in fact, sustainable.

They also mention the Ramsar (Wetlands) Convention, it should be spelled out in full and the 'wise use' clause cited.  That is specifically referring to what we now can sustainability, as is the Biodiversity Convention ("sustainable use' there).  When those are mentioned first, their titles should be spelled out in full.

Without these additions, my own feeling s that the paper would be more-appriately placed in specialized botany or ecology journal such as Wetland Ecology and Management, and not in Sustainability.

Lastly, the Appendix should be in English, since the paper is.  Although many of us can read French, others reading Sustainability may be unable.

Those are my only comments.

Reviewer 2 Report

The work deals with interesting and important issues concerning the biodiversity of wetland habitats of tropical regions. The paper significantly complements the state of the art about studied ecosystems. The introduction was substantive and well justified the research. The methods used in the study were clear and correct.  Material and methods typical for botanical research were used, making the experiment replicable for other researchers. The results were presented in an orderly and clear manner. The discussion largely explained the results obtained and compared them to the existing state of knowledge. The attached list of species is a valuable addition to the results. Substantive work represents a good scientific level. Some aspects need fundamental revision. The hypothesis and purpose of the work were not clearly formulated, so it is difficult to assess whether the conclusions support established assumptions. The literature used in most cases is difficult to access and non-English-language, so it is difficult to assess the adequacy of its selection and relevance to this work. The language of the publication is at a moderate level, but there are editing errors and sections in a language other than English. Descriptions and titles of figures are also missing in many places. The readability of the aforementioned figures should be improved, the form should be standardized and the figures should be adjusted to meet editorial requirements. The writing of the Latin names of plant species should also be standardized, as there are different forms in the text. The text also happens to have repetitions of the same words in sentences, where synonyms could be used, for example. The notation of units and numerals should be standardized according to editorial requirements. It also appears that the text got formatted badly during the conversion from a Word file to a PDF, no less, it cannot remain in this form. The order of chapters on Back Matter appears to be incorrect based on Instructions for Authors. The form of the literature section needs to be adapted to the requirements of the journal. The overall assessment is positive, the work is suitable for publication, however, it needs a major revision.

Specific comments according to the line:

13-14 „erosion control and climate control” can be replaced with other words to avoid repeating the word "control."

25 Latin names of species should be written in italics.

31 There should be no referring to tables and figures or other parts of the paper in the abstract.

42-43 The text is a bit stretched out. Perhaps due to unnecessary spaces or it got formatted that way. Please check in Word.

46 Please arrange the sentences so that the words are not repeated in the same sentence (2 x growth). 

72 Sorindeia warneckei – italics. Please apply to all Latin species names.

80 km2 – superscript

83-84 Units of measurement (meters) are once shown next to the values (100m) and once separately (200 m). Please apply space to all cases ).

90 Unnecessary space.

94 (Figure 1) Figures should have titles. The text should be in English. Figures should be quoted in the text above, before the image appears.

141 These references can be placed in a single bracket: [27,28].

150 The figure reference should be before the period at the end of the sentence.

161-166 Species names of plants should be written in italics.

167 (Figure 2) Figures should have titles. The text should be in English.

175 (Figure 3) Figures should have titles. The text should be in English. Apply to all.

177-179 Translate into English and add to the description of figure 3. As it is done in line 189.

181 Remove text residue from a sentence   „%).”

186 (Figure 4) Add figure title, translate text into English.

189 Unnecessary space at the end of the line. A description can be included in the table caption above.

221, Figure 6. Use English. Align the frames of the figures. It is also a good idea for the scales to have the same range. Figures should have headers, if showing two figures as one, they should be signed as (a) and (b). The reference in line 224 „(figure 6b)” suggests that this division was introduced, however, it must have been lost in the formatting proces, and is not visible in PDF.

222 Once again, unnecessary repetition. The sentence could be phrased: „Distribution of biological and chorological types in riparian forests”. A description of the abbreviations of the habitat names should be added, as has been done under the previous figures.

233 Bambusa vulgaris – italics

246-247 It is not necessary to provide numerals in both written and numerical forms. The generally accepted form is numbers as words in quantities from one to ten (one, two, three...) and numerals above the value of ten (11, 12, 13...). I suggest using only numerals in this case.

248 I believe that (%) is missing from the values in brackets.

251 The drawings should have titles. Designations (a) and (b) should be placed at the top left corner.  Text is not in English. One scale for figures a and b would be preferable.

252 A description of abbreviations should be added.

253-254 Isn't there a part of the sentence missing? Did you mean: Phytogeographically the ponds are dominated by pantropical (46.95%) and Guineo-Congolese (32.44%) species (Figure 7b).?

255 This sentence is a bit odd. Did you mean: …followed by Afro-American (12.50%) and introduced species (10.41%)? Or maybe: ..followed by Afro (12.50%) and introduced -American species (10.41%)?

257 Please be consistent. Once you give values such as fractions (35.41%), other times whole numbers (25%).

267-269 Providing values in numbers alone is sufficient.

272 I think you meant (2.25%) or (22.5%).

278 This is not the right place to quote figure 8. How about at the end of a sentence or after fragment: „(26.52%) species” in line 279?

284 Figure 8 is barely readable. The height of the drawings should be increased. He also proposes to standardise the scale, give a title and introduce explanations of abbreviations in the graph caption.

285 A more correct way to write it: „Distribution of biological and chorological types of reservoirs”

294 Numbers are sufficient.

295-296 „The most frequent species are” I suggest you use the past tense "were".

299 Increasing the height of the figures would improve readability. Move a and b markings to the top. Captions should be in English.

301-302 Something is missing in the sentence, maybe „and”?

303 Afro-tropical species?

309 Please be consistent, if we put a dot after the sp. then we do so in each case. Please apply to the rest of the document. See the lines: 230, 232, 238, 261, 262, 287, 288, 309, and 312.

319 space

320 space

338 The word „and” should not be written in italics.

363 Incorrect citation, sentence ending before brackets.

371 paleo-tropical?

379-382 The font of the text should remain the same size.

381 Did you mean protection (originaly: regulation) and usage regulation?

401-411 The text related to the explanation of abbreviations was badly formatted, this should be corrected.

401-421 It seems to me that supplementary materials should be placed after the discussion and conclusions and before the author contributions and funding information.

421-504 The literature list must be standardized and adjusted to meet editorial requirements. Author 1, A.B.; Author 2, C.D. Title of the article. Abbreviated Journal Name Year, Volume, page range.

Reviewer 3 Report

I like this work that mainly describes plant biodiversity and vegetations composition in a semihumid river-marsh systen in Togo. It is concidering human impact on the landscape and implications for sutsainability - and address lack of mangament sturctures and local knowledge/adminstrative resources to create good management of the area in benefit to both humans and biodiverstity.

I have some minor comments - some is remarks to be handled through the whole manusccipt - others are addressed to the line/lines to be regarded for improvement.

First: The occurances of plant species, genera and plant famlies are given in very excact % - exsample -  12,71 % - and so on. This is of course mathematical correct - but hardly so biologically. I would at least have left the second digit after comma - but this might be an editorial desicion to handle. You could preobably also just leave all decimals after comma - and explain this simplification in methods.

Second: Several places we find species, genera and familiy numbers given, but some times it starts with Families, then genera and so the number of species - other places it start with species  ---. I  suggest  using the same way of presentation all over - and to start with the highest classification and go down to species.

Third: The use of latin names must be controlled - at least the way it is written - and to me it will be natural to give all latin names in "italics". There are a mixture in the text in this matter. And - regarding latin names - it looks like you sometimes put in the Latin names author id - - and most times not. Be concequent !  (And leave it if not really needed for some reason).

Four: Jumping to the litterture list/References: It is nothing wrong - and in this case also natural to have several papers cited that are in French. AS French is not the language handled by a broad spectre of scientists around the world - an English translation of their titles would be welcomed - but again - up to he editors to decide the policy here. It might also be existing as abstracts ?  (French is the official international language in these countries  but better to see  for peolple interested in these publicatons to se what they handle about without going thoruogh translations).

So to some details:

Line 23: Would "established" be better than "evolved" - I think so.

Line 31: Put  Fig. 5 before Axis 1.

Line 72: Species name in Italics

Line 75: Maybe explain "Lama Depression" in some way  -  an area or a pinpointed spot ?

Fig 1: I would have liked to enlarge this to a whole page and make explantions in english text instead of /or in adition to french.

Line 148: Is it correct to use the word "global" here. I would better say "Local"

Line 162: the number of decimals after comma - etc.

LIne 163-166: Here it is both the use of Italics and the species atuthos - as mentioned earlier. - Make it similar all over.

Figure 3 and 4: The concepts Species Brut and Species Pondere - That is for me concepts I cannot find explained - and is not in my normal vocabulary as I am mainly an animal ecologist -- explin in figur text.  Should put in % on the Y-axis. - - Also in other figures when appropriate.

Figur 5. Please explain the axis a litle better

Figure 6: Is the Y-axis the % number of species - make it clear.

Lines 206-219: This is also a general comment, Do we need to have the number of species etc both in letters and numbers ? I advice to  only use numbers. (Now it can be confused as citations, - too).

Line 233: Easy to overlook, but write Bambusa vulgaris in Italics

Lines 249-250: These lines should be rewritten. It gives no meaning that  the Cocus nucifera is 12 % of the species. It cannot be more than one species -??

Fig. 7 aand b: Please control if % is correct between text and figure - as I think the bars for Spectre Brut and Spectre Pandere might be mixed ?

Line 263: The word  " fallow" - I do not quite understand what it is - as I cannot find any relevant translation either in my dictionary. Is it some kind of catle ?? Well - you understand that I do not understand -- so maybe another word is better or an explanation.

Line 272: it must  be 2,25 % - not 225 !

Line 314: the ine - -- reforestation ( A. aureciliformes). Is the area partly reforestated with this species ? Do not catch what is really meant here. 

Line 348: Should pantropical be Pantropical --- ?

Line: 353: The word "enriched" is used here. But I understand the text as it handles abut more or less intruduced or invasive species dure to huamn activity. If so - enriched is not the corect word - it is degraded due to human  intrussion of human activity and in that regard negativ. (I might misunderstand - but if so there might also be need to make it clear).

Line 259: Is the word mutate correct here ? - (would mean changes due to mutations in the species genes) - If it is changes of the floristic composition - then the word Changes should be used.

Line 364 - 368: "Contamination " of the food chain - Try to rewrtite this sentence - I do ot find "contaminaton" to be the best word.

Line 401 - 419. I lokked for anEnglist text for the Appendix - and would havew aske for it - but now I saw it was there in small letters. Maybe possible to edit in another way - smile !

Line 504: 1 - should probably not be here ?

Reviewer 4 Report

This study “Flora and typology of wetlands of Haho river watershed, Togo”was conducted to  analyze the floristic variety and ecological classification in Haho river watershed. Overall, the structure of the article is complete and the content and purpose are clear. However, there are two main problems. 1. I can sense that the authors' taxonomic foundation of plants is not very good, because of the large number of plant Latin non-conformities and the use of unstandardized names from the text. 2. This study is only a basic analysis of the flora, and as the authors themselves suggested, there is really a need for deeper excavation and analysis. However, I think this work is meaningful for the understanding of the composition of regional wetland flora and future research on the conservation of wetland plant diversity. Therefore, I think we can still give support to the publication. Here are some comments are as follows:

1. Line 74-75:  little research has examined the floristic diversity, phytosociology, or production dynamics of wetlands at an eco-floristic or watershed scale. I don't think this kind of research topic belongs to little research has examined. In the introduction part, the author seems to have been introducing the role, importance and protection of wetlands, but ignored the research progress of wetland floristic and the significance of wetland flora research. Therefore, it is recommended to add this part.

2. There are many places in the article that use French vocabulary expressions that I would suggest are uniformly expressed in English, including the diagrams. Because, in fact, these French words give me trouble to read. eg. Figure 1,2,3,4,6,7,8,9.

3. The format of references in the text is inconsistent, such as line 122, 141,144.

4. Line 150: This taxonomic procession is dominated by 150 Poaceae (14.95%) followed by Fabaceae (11.98%),... Here the results are not consistent with those in Figure 2. It must be verified that.

5. Line 159,160: The genus name needs to be italicized.

6. Line 162-166: The genus name here does not conform to the rules for writing botanical Latin names in literature, and the genus name here cannot be abbreviated. And the botanical Latin names should be italicized. Same Q in line 209-211,272,289,29,297,311,338.

7. Line 169: It is recommended to explain micro-phanerophytes/therophytes/ nano-phanerophytes more clearly, such as micropolymers (between 2 and 8 m), and nanopolymers (<2 m but>0.25 m).

8. Line 230,237,239,263,297,309,312: The AND in the botanical name does not need to be italicized.

9. Line 379-383: the Font size is inconsistent.

10. Please indicate in the article the basis of the plant naming, which is recommended to be corrected by the APG IV system. Because the names of some plants listed in the article have been changed. For example: Acacia dudgeoni Craib ex Hall, its accepted name now is Senegalia dudgeonii (Craib) Kyal. & Boatwr. The Acacia polyacantha Willd. campylacantha (Hochst. ex A. Rich.) Brenan, its accepted name now is Senegalia polyacantha subsp. campylacantha (Hochst. ex A.Rich.) Kyal. & Boatwr. Please check all the names of the plants in the paper.

11. In Annexe: The words of subspecies and varieties in the name also need italics. Please check the species number: 4,5,26,49,65,83,99,100,121,130,161,166,229,255,272,278316,318.

Reviewer 5 Report

Thank you for the opportunity to review your manuscript. It is an interesting study describing the flora of the Togo wetlands that may be of interest to an international audience. However, the text needs intensive editing and revision before its eventual publication. 

My general comment is that there is a need for better characterization of the wetlands studied - type, location in the landscape element, hydroperiod, trophic conditions, etc. Without learning these essential characteristics, comparing the presented data with results from other geographic areas will not be easy.

Please revise the Introduction section, remove unnecessary, commonplace textbook descriptions of the role and function of wetlands and focus on the substance of the manuscript: floristic diversity of wetlands - factors controlling biodiversity - results of similar studies from this, or comparable biogeographic regions. 

The authors probably exaggerate when they write that the material presented contributes to an understanding of wetlands in Togo. The manuscript does not deliver causal studies or in-depth ecological analyses.

The caption of Fig. 1 is flawed. The figure does not show the location of the wetlands studied.

Please make a correct characterization of the studied wetlands, using, for example, Ramsar wetland type classification.

 Please thoroughly describe the purpose and methods of statistical analysis.  

Please change the French axis captions and figure legends to English

Line 194-200 interpretation of PCA results need to be clarified; on what basis did the authors determine gradients of change in species occurrence?  

 What do the authors mean by "edaphic humidity"?

Unfortunately, the conclusions do not fully follow the presented data and need improvement. 

Round 2

Reviewer 2 Report

I am pleased to say that the manuscript has been significantly improved. Certain areas need further revision.

1. The abstract should be shortened to 200 words, at the moment it is over 300. I do not see the need to provide such detailed methodology in the abstract.

2. The writing of Latin plant names should be standardised. If the name appears in the text for the first time, the full species name should be given in italics plus an abbreviation of the discoverer's name. If the name appears for the second time, it can be abbreviated. However, I see that abbreviated and full names often occur next to each other. For the sake of uniformity, I suggest that you write the full Latin name plus the discoverer in each case. You don't use species names that often, and you have the full correct names ready in the appendix. Use them in the text. 

3. Add descriptions under Figures 6-9. most abbreviations for biological forms (Figures 6a-9a: mPh, Th, nph, He, Ge, Lnph, LTh, LHe) and biogeographical types (Figures 6b-9b: Pan, GC, SZ, AT, Am, Aas, S, G, Cos) are not unfolded either in the methodology or in the results. Only the most dominant forms listed in the text can be identified. I suggest adding a description of all forms found. If they are shown in the diagram, they should be explained by its description. Annex L406-415 is a good example.

(4) References and citations should be put in order. If new items are introduced in the middle of the text, unfortunately the numbering of the references must change.

L 12-33 The abstract is too long, it should be up to 200 words.

L 41-42 There is no point in inserting a citation in the middle of a sentence. It is better to do it at the end in the right order: [2,3].

L 49 You can cite like this: [4-7].

L 50-52 I suggest quoting the items at the end of the sentence: [8-10]. According to the authors' guidelines, square brackets with item numbers should be placed before punctuation, i.e. before a dot at the end of a sentence or possibly before a comma in the middle of a compound sentence.

L 71 The full scientific name of Sorindeia warneckei Engl. should be given.

L 79-80 Figure 1 should be quoted at the end of the sentence.

L 89 Please provide full species names.

L 89 Antiaris africana - this species is given as one of the dominant species and is not in the appendix. It is a synonym of Antiaris toxicaria Lesch. ssp. welwitschii (Engl.) C.C.Berg, however please decide which name you are using.

L 121 Item [44] should not come before the earlier items [22,23]. In this case, it should be given the number [22] and the numbers of the following items should be changed. Changes should also be made in the list of references.

L 123 space

L 142 Similar situation. Item [45] has been added to the text before item [29]. The references should be re-numbered in the order of its first occurrence in the text.

L 142-143 Missing punctuation marks in sentences.

L 147 The figure should be quoted before the dot at the end of the sentence.

L 165 Replace French „é” with English „e” (Fréquency).

L 176-179 This text should be attached to the description of Figure 3 instead of the main text. In the case of figure 4 it is correct.

L 182 A fragment of a sentence is missing.

L 207-209 Latin names of species are written in italics.

L 219 There are abbreviations in the diagram that require explanation. Add an explanatory note for the abbreviations in the diagram description, as done under figure 4.

L 225-236 Entire Latin names should be written in italics, including sp.

L 248 Figure 7: Explanation of abbreviations needed.

L 252 African-American species?

L 268 Italics.

L 269 A. polyacantha Willd. This form is not correct. Either write the full name plus the discoverer, or just the abbreviated name A. polyacantha. I suggest using the full names everywhere for consistency.

L279 Add description of abbreviations.

L289 t21?

309-313 Unnecessary spaces, visible gaps in text.

317 'and' does not need to be in italics.

L 394-405 and 406-415 Content essentially repeats itself. I suggest cutting out section 394-405 and moving it under figures 6-9 where it will be useful.

Author Response

Reviwer2# second round comments

  1. The abstract has been shortened to 192 words.
  2. The writing of Latin plant names has been standardized
  3. We Add descriptions under Figures 6-9.

(4) References and citations were put in order.

L 12-33 The abstract has been already shortened.

L 41-42 citation put at the end.

L 49 citation reordered

L 50-52 quotation put at the end.

L 71 The full scientific name of Sorindeia warneckei Engl given

L 79-80 Figure 1 quoted at the end of the sentence.

L 89 full species names provided.

L 89 well noted.

L 121 citation reorded.

L 123 space

L 142 Similar situation. Item reordered.

L 142-143 Missing punctuation marks in sentences repalced.

L 147 The figure is now quoted before the dot at the end of the sentence.

L 165 French „é” with English „e” replaced.

L 176-179 corrected.

L 182 A fragment completed.

L 207-209 Latin names of species are written in italics, corrected.

L 219 There are abbreviations in the diagram with explanatory note for the abbreviations in the diagram description, as done under figure 4.

L 225-236 Entire Latin names written in italics.

L 248 Figure 7: Explanation of abbreviations added

L 252 African-American species?

L 268 Italics.

L 269 suggestion took into account.

L279 Description of abbreviations added.

L289 t21? Don’t understand what this means.

309-313 Unnecessary spaces deleted.

317 transformed.

L 394-405 and 406-415 corrected

Reviewer 5 Report

Dear authors,

Despite your undoubted contribution to improving your manuscript, I still see neither a new version of the text nor a cover letter with highlighted corrections addressing the reviewer's comments. The text continues to be riddled with spelling errors - e.g., "Arthimetic " in the legend of Figure 8. 

Thus, I cannot consider the manuscript corrected enough to be accepted for publication.

Author Response

We appreciate the reviewer's insightful comments and ideas for enhancing the quality of the article.

The reviewer's proposed characterization of the wetland was mentioned as requiring additional research to delve deeper into the topic, but the focus of our current research is the floristic diversity.

The introduction has been changed.

The phrase has been modified.

The number has been revised.

Consideration was given to characterization suggestions.

The goal and methods of statistical analysis were to demonstrate the floristic diversity and distribution of species over the study area.

French axis labels and figure legends have been translated into English.

Changed

Edaphic humidity is soil moisture.

Improved conclusion

Round 3

Reviewer 5 Report

I appreciate the effort to improve the manuscript. Please remove minor deficiencies such as :

line 93 - please remove the letter V

line 127 - what is the phrase "sigmatist approach" supposed to mean?

What is the role of the regression line equation in Fig. 2?

Please re-read critically and edit the manuscript if necessary